# Conformational changes in the human Cx43/GJA1 gap junction channel visualized using cryo-EM

Hyuk-Joon Lee [1,4], Hyung Jin Cha [1,4], Hyeongseop Jeong [1,2,4], Seu-Na Lee [1], Chang-Won Lee[1], Minsoo Kim[3], Jejoong Yoo[3] & Jae-Sung Woo [1] ✉

Connexin family proteins assemble into hexameric hemichannels in the cell membrane. The hemichannels dock together between two adjacent membranes to form gap junction intercellular channels (GJIChs). We report the cryo-electron microscopy structures of Cx43 GJICh, revealing the dynamic equilibrium state of various channel conformations in detergents and lipid nanodiscs. We identify three different N-terminal helix conformations of Cx43—gate-covering (GCN), pore-lining (PLN), and flexible intermediate (FIN)—that are randomly distributed in purified GJICh particles. The conformational equilibrium shifts to GCN by cholesteryl hemisuccinates and to PLN by C-terminal truncations and at varying pH. While GJIChs that mainly comprise GCN protomers are occluded by lipids, those containing conformationally heterogeneous protomers show markedly different pore sizes. We observe an α-to-π-helix transition in the first transmembrane helix, which creates a side opening to the membrane in the FIN and PLN conformations. This study provides basic structural information to understand the mechanisms of action and regulation of Cx43 GJICh.

Gap junction intercellular channels (GJIChs) facilitate direct communication between adjacent cells and play important roles in various biological processes such as cardiac contraction, electrical coupling, cell differentiation, and death[1]. GJIChs are formed when two opposing hemichannels (also called connexons) from adjacent cells are docked[2–8], each being composed of six connexin (Cx) protomers[9]. In humans, 21 connexin genes have been identified and categorized into classes A–E based on similarities between amino acid sequences. Compared with other membrane channels, GJIChs have relatively large pores that allow the passage of ions and metabolites and enable intercellular communication over a large area in various tissues[10,11].

Cx43, also known as GJA1, is the most ubiquitously expressed connexin protein. It is detected in most cell types and involved in many biological processes. For example, it is the most abundant connexin in cardiomyocytes, bone cells, skin cells, and astrocytes and plays crucial

roles in synchronizing the contractions of the heart[12], regulating the bone mass via mechanotransduction[13], maintaining the integrity of the avascular epidermis[14], and coordinating the activities of the central nervous system[15]. Mutations or the misregulation of *Cx43* can cause many human diseases such as oculodentodigital dysplasia (ODDD), palmoplantar keratoderma, heart diseases, and cancers[16,17].

The gating and permeability of Cx43 GJICh are regulated by many factors such as transjunctional voltage (Vj), pH, divalent ions, interacting proteins, reactive oxygen/nitrogen species, phosphorylation, and membrane lipids[18–20]. Although the gating mechanism of Cx43 GJICh is not yet clearly understood, mutational studies have suggested that its N-terminal helix (NTH) is crucial for channel gating and charge selectivity. Mutations in two conserved residues (W4A and ODDD-linked L7V) in the NTH greatly diminish the dye transfer activity and junctional conductance of Cx43 GJICh[21]. A double mutation of D12S

¹Department of Life Sciences, Korea University, Seoul 02841, Korea. ²Center for Research Equipment, Korea Basic Science Institute, Chungcheongbuk-do 28119, Korea. ³Department of Physics, Sungkyunkwan University, Suwon 16419, Korea. ⁴These authors contributed equally: Hyuk-Joon Lee, Hyung Jin Cha, Hyeongseop Jeong. ✉e-mail: jaesungwoo@korea.ac.kr

and K13G almost completely removes the Vj-dependent gating property of Cx43 GJICh[22]. The cytoplasmic loop (CL) with approximately 40 amino acids and the C-terminal domain (CTD) with approximately 150 amino acids also play important roles in the gating regulation[18]. While Cx43 GJICh exhibits the maximum $H^+$ transmission at a pH of approximately 6.9, the transmission activity reduces sharply with decreasing or increasing pH until the channel closure occurs at pH 6.4 or 7.6[23,24]. A Cx43 derivative with the deletion of residues from K258 to the C-terminus (Cx43-M257) has been found to be pH-insensitive and shows the maximum activity in the pH range 6.4–7.6[24–26]. The C-terminal deletion mutation removes the residual state, increases the mean open time, and slows the transition between the open and closed states[27].

The low-resolution structure of Cx43 GJICh was determined using electron cryo-crystallography in 1999[4,28]. This provided structural insights into the packing arrangement of 12 transmembrane domains (TMDs) and the docking of two hemichannels. The crystal structure of Cx26 GJICh reported in 2009[5] revealed an overall dodecameric architecture and detailed intermolecular interactions between connexin protomers composed of NTHs, TMDs, and extracellular loops (ECLs). Although the interactions between NTHs and TMDs were not clear in this structure owing to the weak electron densities of NTHs, recent cryo-electron microscopy (EM) structures of Cx46/Cx50 GJICh[7] have shown that amphipathic NTHs line the pores through hydrophobic interactions with the first transmembrane helix (TM1s) and TM2s, resulting in a hydrophilic pore pathway. This structure has therefore been regarded as the open state of GJIChs.

The completely closed conformations of GJIChs have not been determined at high resolution. Although several low-resolution structures of Cx26 GJIChs closed by the M34A mutation or acidic pH have shown a large density blob of NTHs blocking the pore pathway[29–31], more evidence is needed to clarify whether NTHs function as physical plugs. Recently, we determined the cryo-EM structure of the Cx31.3 hemichannel[9], which showed a narrow pore (approximately 8 Å in diameter) formed by six NTHs horizontally covering the cytoplasmic gate. However, it remains unclear whether GJIChs and other hemichannels also adopt this NTH conformation to downregulate channel activity.

In this study, we conduct cryo-EM single particle analyses of Cx43 GJICh under various conditions that affect the channel activity. The analyses reveal four different conformations of the Cx43 protomer and their distribution in GJICh particles. We also find the detailed interactions of the channel with lipids/detergents. This study provides high-resolution information on dynamic structural changes in Cx43 GJICh, which would be necessary to understand the mechanisms of gating, permeability, or regulation of the channel.

## Results

### Sample preparation and structural determination of Cx43 GJICh

To understand the gating mechanism of Cx43 GJICh, we performed a cryo-EM study on the channel under eight different environmental or mutational conditions and produced 12 reconstruction density maps (conditions 1 to 8; Fig. 1a–c, Supplementary Figs. 1a–j and 2, and Supplementary Table 1). In all of the conditions, except for the glycodiosgenin (GDN) detergent environment (condition 4; Fig. 1c and Supplementary Fig. 1c, d), human wild-type Cx43 (Cx43-WT) and its mutant (Cx43-M257) GJIChs were purified using lauryl maltose neopentyl glycol (LMNG) and cholesteryl hemisuccinate (CHS) at a 10:1 ratio by mass (hereafter referred to as LMNG/CHS; Fig. 1c and Supplementary Fig. 1a, b, e–j). Although dodecameric GJIChs were partly dissociated during purification in LMNG detergents without CHS, they were mostly stable in GDN detergents. Hence, we could completely exclude LMNG and CHS in condition 4. All protein samples were purified at pH 8.0, however, for the experiment at a pH of 6.9 (condition 3), we lowered the pH to approximately 6.9 prior to vitrification by adding a 500-mM HEPES buffer at pH 6.8.

To assess the structures in the lipid environments, purified channels in LMNG/CHS were reconstituted into lipid nanodiscs containing soybean phospholipids or 1-palmitoyl-2-oleoyl-phosphatidylethanolamine (POPE). In conditions 5–7, phospholipids were solubilized with approximately 4-fold more LMNG/CHS by mass before the reconstitution process. A high amount of CHS was therefore likely included in the final lipid nanodisc samples (Fig. 1c and Supplementary Fig. 1e–h). However, because phospholipids were solubilized with only LMNG under condition 8, the CHS content in this sample was likely very low (see Methods; Fig. 1c and Supplementary Fig. 1i, j).

In the cryo-EM image processing of the eight datasets, a three-dimensional (3D) classification showed a single conformation for five conditions (1, 4, 5, 6, and 7; Fig. 1d, left) and multiple or ambiguous NTH conformations for three conditions (2, 3, and 8). For the latter three datasets, we further performed protomer-focused analyses to identify the distribution of individual protomer conformations in GJICh particles (Supplementary Fig. 3a, b) and/or hemichannel-focused analyses rather than GJICh-focused ones to classify GJICh particles more precisely and reconstruct more homogeneous maps with higher resolutions (see below; Fig. 1d, right, e and Supplementary Fig. 3c). This resulted in the identification of four different protomer structures (Fig. 1e and Supplementary Fig. 3b, c). Many undocked hemichannels were found in all datasets, but they were not further processed due to preferred particle orientation.

All the structural models in this study contained NTHs, TMDs, and ECLs, but not CLs and CTDs, which remained unresolved owing to poor EM density (Fig. 1a, b). Moreover, because the map density for the N-terminal methionine (M1) was completely invisible, we investigated N-terminal modifications using a mass spectrometric analysis. Met1 was not detected, and the following G2 residue was partially acetylated (Supplementary Fig. 1k). However, we could not identify the acetylation state of G2 from the cryo-EM data because of the ambiguous map density of this position. We, therefore, included non-acetylated G2 in our final models. In all structures, we identified strong map densities of three disulfide bonds in ECLs, indicating that the presence of 2 mM $\beta$-mercaptoethanol in the protein samples did not substantially affect the ECL structures (Supplementary Fig. 1l).

### Structures of Cx43 GJICh in the gate-covering NTH (GCN) conformation

Since Cx43 GJIChs in conditions 1, 4, 5, 6, and 7 were conformationally homogeneous, we determined a single structure for each condition at resolutions 2.4–3.6 Å (Fig. 1d and Supplementary Fig. 4). All five structures showed an identical conformation, with six NTHs in each hemichannel horizontally covering the channel gate, which we referred to as the full GCN conformation (Fig. 2a, b). Although this conformation was different from the full pore-lining NTH (PLN) conformation of previous Cx26 and Cx46/50 GJICh structures (Supplementary Fig. 5a), many structural features were consistent with those of previous structures. For example, the resolution of the extracellular region was higher than that of the cytoplasmic region (Supplementary Fig. 4); water molecule densities were mainly identified in the extracellular region (Fig. 2a, b); and acyl chain densities surrounding the TMDs were more ordered in the outer leaflet than in the inner leaflet (Fig. 2a, b).

Because NTHs are located inside the GJICh pore, they always form a constriction site in the pore region with the smallest diameter. In the full GCN structure of Cx43 GJICh, the constriction diameter determined by the six N-termini facing toward the pore center had a solvent-accessible diameter of approximately 5.0 Å (Fig. 2b), which was smaller than the hydrodynamic diameter of ATP (approximately 9.8 Å)[32] and the diameters of the constricted pore regions in the available hemichannel or GJICh structures. Therefore, this conformation likely represented a closed state for large metabolites such as ATP. However, it should be noted that, since the central pore is not completely closed

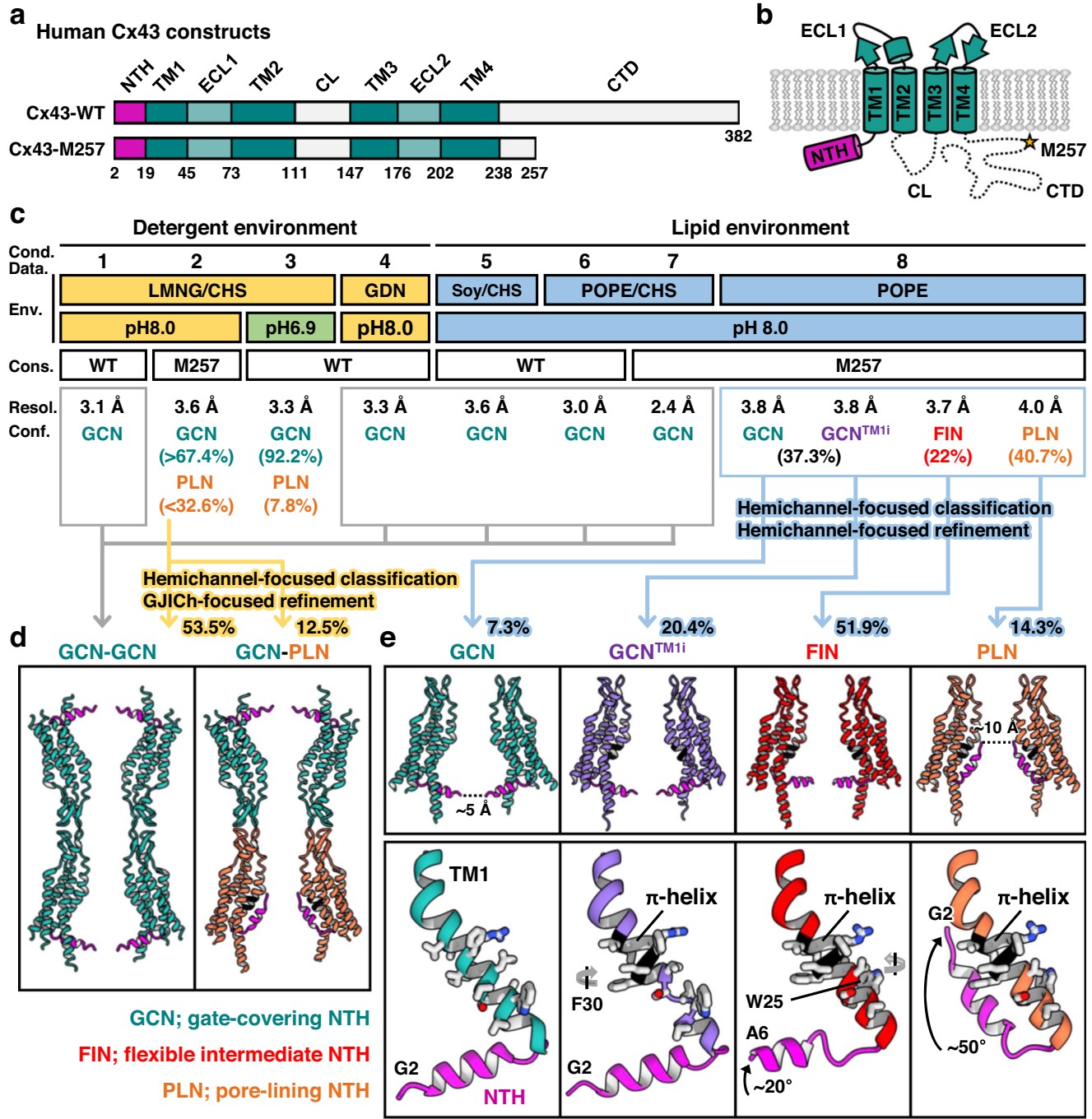

**Fig. 1 | Overview of the purification and structural determination of Cx43 GJICh. a** Domain architectures of Cx43-WT and Cx43-M257. **b** Topology diagram of Cx43. The dashed lines indicate flexible loops that could not be structurally determined. The M257 residue is marked and labeled. **c** Purification conditions for Cx43-WT and Cx43-M257. The final buffer conditions for Cx43 proteins are indicated, such as pH, detergents, and lipid composition. The percentages of the GCN and PLN protomers in parentheses are indicated as per the promoter-focused analyses. See the Methods section for details. Abbreviations: cond., condition; data, dataset; env., environment; cons., construct; resol., resolution; conf., conformation. **d** Ribbon presentation of the conformationally homotypic (left; conditions 1, 2, 4, 5, 6, and 7) and heterotypic GJIChs (right; condition 2), as viewed from the membrane. For clarity, only two protomers per hemichannel facing each other are shown. The NTHs and π-helices are colored in magenta and black, respectively. **e** Ribbon representation of four different hemichannel sub-structures of Cx43-M257 GJICh obtained using hemichannel-focused processing, as viewed from the membrane (top; condition 8). Only two protomers facing each other are shown. Conformational changes in NTH and TM1 of protomers in the four conformations (bottom). Movements of the NTH are indicated by one-way arrows. Rotations in the W25 and F30 residues are also indicated.

and the N-terminal residue (G2) is likely flexible, atomic ions would be able to pass through the pore center.

The full GCN structure of Cx43 GJICh was highly similar to that of the Cx31.3 hemichannel structure (Supplementary Fig. 5b, c). The conformation of each NTH was mainly stabilized by intramolecular interactions with TM2; L10 and V14 of NTH participated in

intramolecular hydrophobic interactions with Y92, L93, V96, F97, and M100 of TM2 (Fig. 2c). These residues are hydrophobically conserved (>80%) in connexins A and C, supporting the hypothesis that GJIChs in classes A and C have similar GCN conformations (Fig. 2d)[9]. The intermolecular interactions included a salt bridge between D12 and R101 and a long-distance hydrophobic interaction between L11 and F97

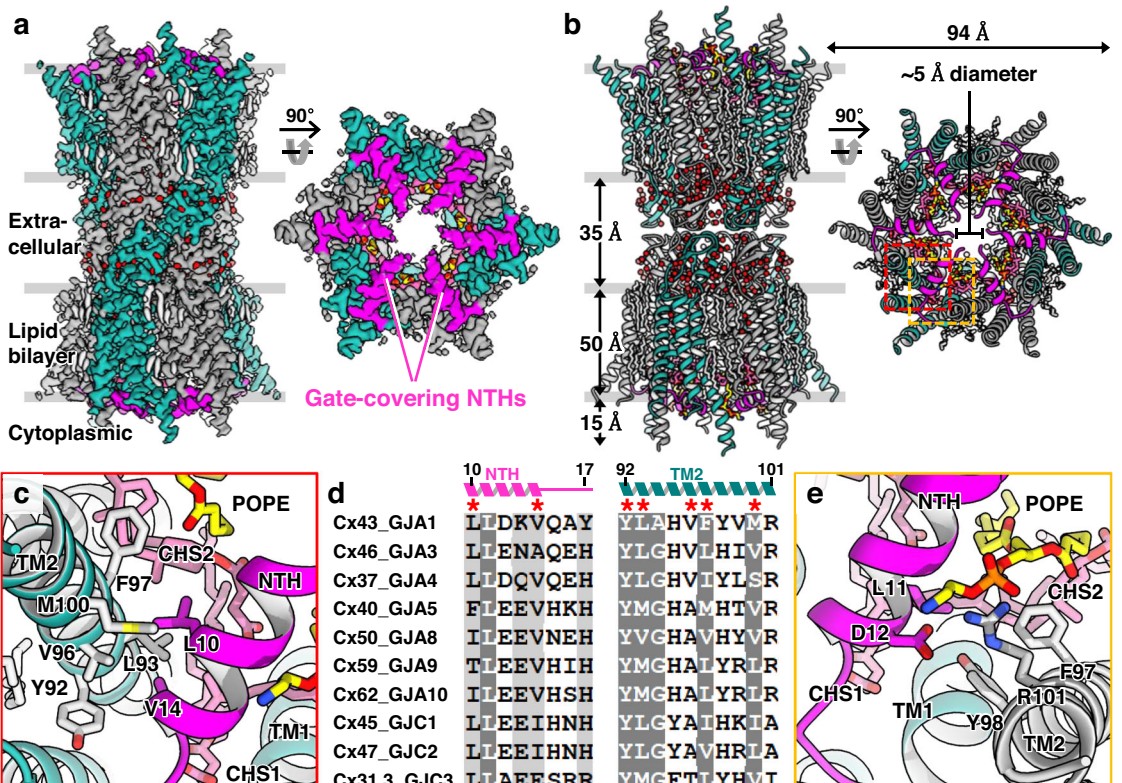

**Fig. 2 | Structure of Cx43-M257 GJICh in GCN conformation. a** Cryo-EM map density of Cx43-M257 GJICh in POPE/CHS (condition 7) viewed from the membrane plane (left) and cytoplasmic side (right). The NTHs are colored magenta. Water molecules are shown as red spheres. **b** Ribbon representation of the structure in (**a**). **c** Detailed intramolecular hydrophobic interactions between NTH and TM2. The residues involved in these interactions are drawn as sticks and labeled. **d** Sequence alignment of A- and C-class connexins. More than 90 and 80% similarly conserved residues are shaded in dark and light gray, respectively. The residues involved in the intramolecular NTH–TM2 interactions are indicated by asterisks. **e** Detailed intermolecular interactions between NTH and TM2 in a neighboring protomer. NTH and TM2 in the neighboring protomer are colored magenta and gray, respectively.

(Fig. 2e). We did not find any significant interactions between neighboring NTHs.

## CHS and phospholipids stabilize the full GCN structure

Because the full GCN Cx43 GJICh structure in lipid nanodiscs that mainly contain POPE and CHS (condition 7) was assessed at a high resolution of 2.4 Å (Supplementary Fig. 4), we could clearly identify POPE and CHS molecules and understand their detailed interactions with the channel.

First, we found many ordered acyl chains (gray space-filling models; Fig. 3a) bound not only to the channel exterior but also to the interior. As amphipathic NTHs are dissociated from TMDs, six TMDs in each hemichannel exposed a large hydrophobic surface area in the channel interior (Fig. 3b). The extracellular half of this surface area was almost completely covered with 12 acyl chains. Because NTHs in the PLN conformation specifically interact with the acyl-binding sites, the competitive binding of POPE molecules would substantially inhibit the PLN conformation and stabilize the GCN conformation. Similar acyl-binding modes were observed in the LMNG/CHS structure (condition 1; Supplementary Fig. 5f, g).

Second, we identified densities consistent with two CHS molecules bound to each protomer in the hydrophobic interior of the channel (pink space-filling models or sticks, referred to as CHS1 and CHS2; Fig. 3b–d). CHS1 was located in a deep hydrophobic pocket formed between NTH, TM1, and TM2 (Fig. 3c). While the sterol ring of CHS1 interacted closely with L26, F30, T89, and L93, its hydrocarbon tail interacted with L90. CHS2 was located between CHS1 and NTHs (Fig. 3d). The sterol ring of CHS2 closely interacted with F97 and Y98 and participated in long-distance interactions (4.6–5.3 Å) with the

hydrocarbon tail of CHS1 and two NTHs. The density of the succinyl group of CHS was observed for neither CHS1 nor CHS2, indicating that the succinyl group does not specifically interact with the protein. Therefore, cholesterol likely binds to channels at the same binding sites with similar affinities. The structure in GDN detergents (condition 4) also showed a strong map density at the CHS1-binding site, which has been presumed to be part of the sterol-like ring of GDN (Supplementary Fig. 5h). This finding suggests that this binding site is not highly specific to CHS but may have broad specificity for sterol-like molecules.

Third, we identified a strong map density of POPE in each hole between neighboring NTHs in which both the hydrophobic tails and hydrophilic head of POPE were able to be modeled (Fig. 3e). The lipid tails closely interacted with W4, L7, L11, and F97 of Cx43 within 5 Å and with CHS2, completely obstructing the hole (Fig. 3e). In the lipid head, the phosphate group formed a salt bridge with R101 and interacted with S5 (Fig. 3e). We excluded the possibility that the lipid density originated from potential LMNG impurities as it did not fit LMNG, especially in the maltosyl group, and differed from the corresponding density in the structure of LMNG/CHS (condition 1; Supplementary Fig. 5e). In addition, the guanidino group of R101 moved away from the head group in the LMNG/CHS structure (Supplementary Fig. 5e).

Collectively, these structural analyses strongly suggest that CHS and POPE molecules bind to NTHs, TM1s, and/or TM2s in Cx43 GJICh to almost completely mask solvent-exposed hydrophobic surfaces and stabilize the conformation of the GCN.

## The full GCN Cx43 GJICh is occluded by phospholipids and CHS

Because the hydrophobic pore region formed by TM1 and TM2 helices (referred to as the transmembrane pore region) was almost completely

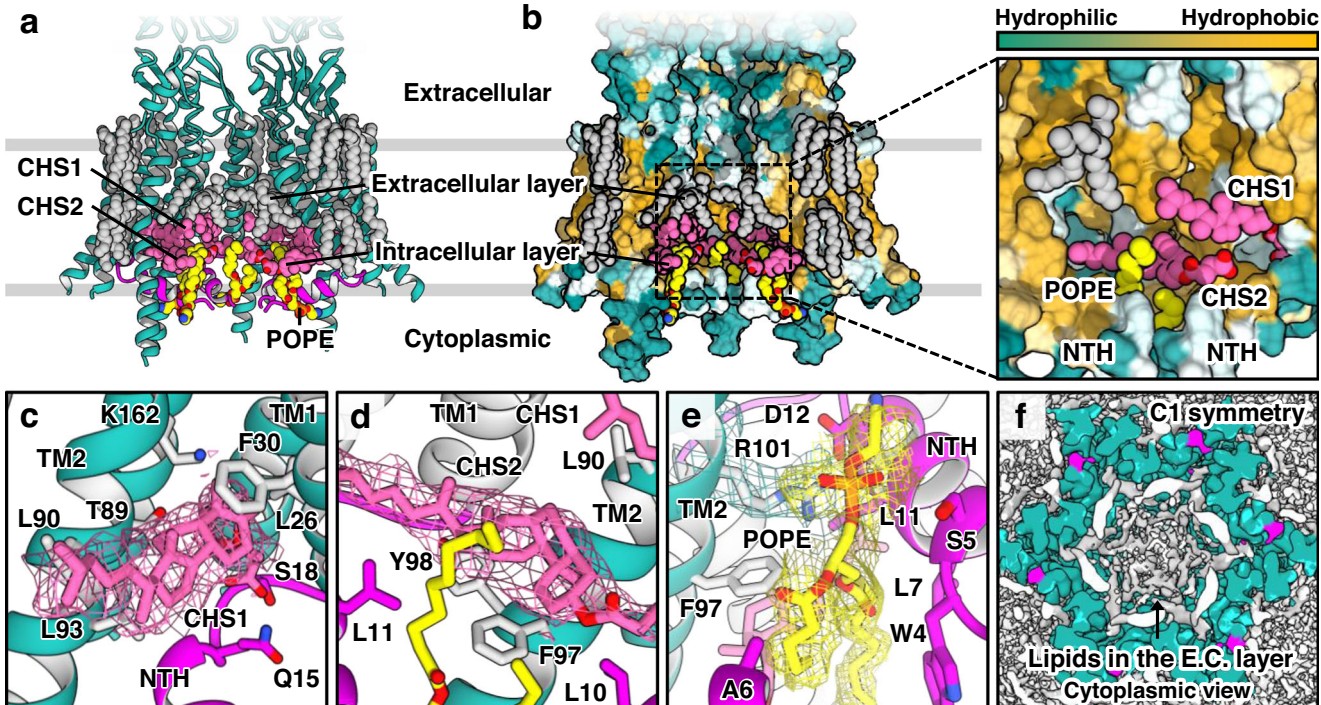

**Fig. 3 | CHS and phospholipids stabilize the full GCN structure. a** Ribbon representation of Cx43-M257 GJICh in POPE/CHS (condition 7) viewed from the membrane. For clarity, only four protomers are shown. Lipids are represented as space-filling models: acyl chains (gray), CHS (pink), and POPE (yellow). **b** The surface hydrophobicity (left) of the structure shown in (**a**) and close-up view of the pocket inside the pore (right). The surface was colored following the hydrophobicity scheme in Chimera. **c**–**e** Close-up views of map densities of CHS1 (**c**), CHS2 (**d**), and POPE (**e**) in the structure shown in (**a**). Lipid molecules and their interacting residues are represented by sticks. **f** Sliced views of the 3D density map processed with C1 symmetry imposition viewed from the cytoplasmic side. Pore-occluding lipids in the extracellular layer (E.C. layer) are indicated. The map contour level was 2.4σ.

covered by POPE and CHS molecules in the Cx43 GJICh structure under condition 7, the empty space inside should have been filled with hydrophobic or amphipathic molecules, although their map densities would not have been clearly seen in the structure. To address this hypothesis and exclude artificial densities generated by D6 symmetry imposed during the 3D reconstruction process, we determined the structure with the same dataset without symmetry imposition (C1 symmetry) at a resolution of 2.7 Å (Supplementary Fig. 4).

In the structure processed with C1 symmetry, we found that many density blobs formed a thin layer at the extracellular end (referred to as the extracellular layer) of the transmembrane pore region in each hemichannel part (Fig. 3f). To determine whether these were noise densities, we compared the densities inside the channel with those of the lipids in the nanodisc and outside the channel (Supplementary Fig. 6a). When we gradually decreased the map contour level, we found that the inner densities appeared slightly earlier than the outer densities and that their sizes grew at almost the same rate. Outside the channel, the densities of the lipid heads were much stronger than those of the tails. This is usually observed in membrane protein structures in lipid nanodiscs, indicating that the heads are more ordered than the tails and/or display enhanced electron scattering by the composition of heavier atoms. Because the inner densities were located immediately next to the heads of the lipids bound to the channel interior and were as strong as the densities of the lipid heads outside, they were likely POPE heads (Supplementary Fig. 6b). Therefore, the channel was likely occluded by POPE molecules at the extracellular end of the transmembrane pore region.

By contrast, although we observed another layer of density blobs at the intracellular end (referred to as the intracellular layer) of the transmembrane pore region, the density blobs were weaker and shorter than those in the extracellular layer at the same map contour level (Supplementary Fig. 6a). This suggests that phospholipids in the intracellular layer may not be highly ordered probably due to six NTHs, and thus the cytoplasmic end of the central pore formed by six N-termini may not be completely closed by lipids.

Strong density blobs in the extracellular layer of the transmembrane pore region were also observed in all the Cx43 GJICh structures in the GCN conformation. However, the densities were relatively weaker in the structures of LMNG/CHS (Supplementary Fig. 6a) than those of the surrounding detergents in each structure. LMNGs may be less ordered because of their heads, which are much larger than those of POPEs, especially in funnel-shaped vestibules.

## Conformational equilibrium of Cx43-M257 GJIChs in phospholipids

Since CHS appeared to contribute substantially to the GCN conformation of Cx43 GJICh, we assessed whether the structure would change in phospholipids without CHS (condition 8). To reduce CHS in the lipid nanodisc sample, we dissolved POPE lipids in LMNG detergents without CHS and mixed them with purified channels in LMNG/CHS for nanodisc reconstitution (Fig. 1c and Supplementary Fig. 1i, j). Because the protein sample included only 0.0005% (w/v) CHS, the lipid-protein mixture contained approximately 300 fold more phospholipid molecules than CHS molecules. In addition, CHS as well as LMNG would be partly removed during the incubation with adsorbent beads. Therefore, the CHS content in the lipid nanodisc sample was likely less than 0.3 mol%, which should be much lower than that under condition 7 (less than 50 mol%).

The protomer-focused conformational variation analysis of Cx43-M257 GJICh particles in the POPE nanodiscs (condition 8) showed six protomer classes in three distinct conformations designated as GCN, flexible intermediate NTH (FIN), and PLN, which accounted for 37.3%, 22%, and 40.7% protomers, respectively (Supplementary Fig. 3a and Supplementary Movie 1). In the FIN conformation, the density

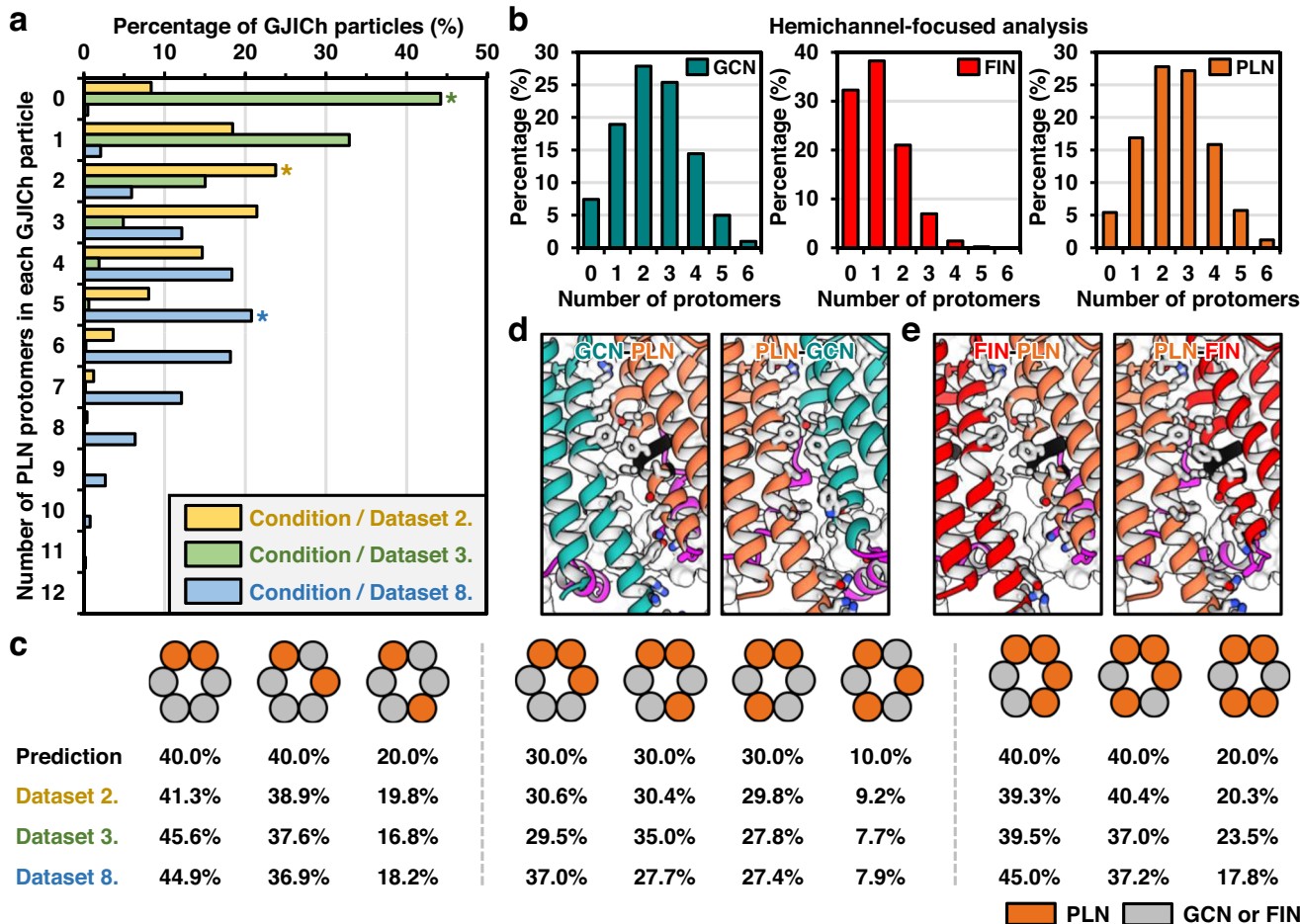

**Fig. 4 | Distribution of different protomer conformations in Cx43 GJICh particles. a** Distribution of various compositions of PLN protomers within GJICh particles. Three datasets of Cx43 in detergents and nanodiscs were analyzed. For each dataset, the highest percentage is indicated by an asterisk. **b** Distribution of various compositions of GCN (left), FIN (middle), and PLN (right) protomers in each half of the GJICh particles for Cx43-M257 in POPE nanodiscs (condition 8). **c** Schematic drawings for the possible arrangements of PLN protomers in hemichannels for the three datasets shown in (**a**). Orange and gray circles represent PLN protomers and the others (GCN or FIN), respectively. For all arrangements, the predicted probability and experimental values from the three datasets are indicated. **d** Interfaces between protomers for Cx43-M257 heteromeric GJICh composed of PLN (orange) and GCN (green) protomers. Residues at the interfaces between the protomers were drawn as sticks. **e** Interfaces between protomers for Cx43-M257 heteromeric GJICh composed of PLN (orange) and FIN (red) protomers.

corresponding to NTH was slightly shorter than that of GCN and PLN and was more kinked toward TMD than toward GCN (Supplementary Fig. 3b).

Next, we traced all the protomers in the PLN conformation (PLN protomers) to their original GJICh particles and examined the distribution of PLN protomers in each GJICh particle. The results showed a bell-shaped (normal) distribution with a peak at five protomers (Fig. 4a). The same analyses for the FIN and GCN protomers showed normal distributions with peaks at two and five protomers, respectively. In total, for 124,079 GJICh particles, we found few GJIChs with all 12 NTHs in a single conformation: 42 fully GCN, no fully FIN, and 42 fully PLN GJIChs. We also analyzed the distribution in the hemichannel region, which showed normal distributions for all three conformations (Fig. 4b).

To determine whether protomers affected the conformations of the neighboring protomers, we analyzed the positional distribution of PLN protomers relative to other protomers in the hemichannel region and compared them with the predicted values when they were randomly distributed. For ease of understanding, three hemichannel groups, each containing two, three, and four PLN protomers, were analyzed separately. For all the possible relative positions of the PLN protomers in each group, we counted the number of hemichannels and compared them with the

predicted values (Fig. 4c). In all three analyses, the experimental values did not differ significantly from the predicted values, although the percentages of two, three, and four consecutive PLN protomers were slightly higher than the predicted values. We also confirmed no steric hindrance between the three different conformations when they neighbored each other after building artificial models of heteromeric channels (see below; Fig. 4d, e). These data strongly suggest that PLN, FIN, and GCN protomers are almost randomly distributed in Cx43-M257 GJIChs in POPE nanodiscs. Therefore, we concluded that our cryo-EM experiment captured the dynamic equilibrium state in which individual protomers freely changed their NTH conformations in a single GJICh.

**The conformational equilibrium of Cx43 GJIChs in LMNG/CHS**

In the protomer-focused analysis with the dataset in condition 2 (dataset 2), we found that three classes (classes 3, 4, and 8) containing 32.6% of protomers were in the PLN conformation, whereas the remaining five were in the GCN conformation (Supplementary Fig. 3a). Therefore, Cx43-M257 GJIChs in LMNG/CHS, similar to POPE nanodiscs, were likely in the conformational equilibrium state of individual protomers between GCN and PLN. Since the environmental conditions between conditions 1 and 2 were identical, the PLN protomers in condition 2 were likely caused by the partial deletion of CTD

(Fig. 1a–c), suggesting that the intact CTD partly contributed to the GCN conformation.

An analysis of the number of PLN protomers in the GJICh particles under condition 2 showed a normal distribution, with a peak at 2–3 protomers (Fig. 4a). The positional distribution of each hemichannel was also close to a random distribution (Fig. 4c). It should be noted that two protomer classes (classes 4 and 8) had weak but detectable map densities of the GCN conformation (Supplementary Fig. 3a). Therefore, the actual number of PLN protomers in the 3D classification data should have been less than 32.6%, and the distribution data of PLN protomers should have contained substantial errors.

A dynamic conformational equilibrium was also observed in Cx43-WT GJIChs (condition 3). We identified PLN protomers (class 2; 7.8%) and their random distribution in Cx43-WT GJIChs at a pH of approximately 6.9 (Fig. 4c and Supplementary Fig. 3a). This suggests that a pH of approximately 6.9 slightly shifted the conformational equilibrium of individual protomers to the PLN state, although the effect was lower than that of the partial deletion of the CTD (Fig. 4a and Supplementary Fig. 3a). However, this equilibrium state of Cx43-WT GJIChs in condition 3 would not represent the maximally open state of the channels at pH 6.9 in the previous electrophysiological experiments[24]. The effect of pH 6.9 would have been largely compromised by high CHS content that pushed the equilibrium to GCN. Therefore, the effect of pH on the channel structure needs to be further investigated in a phospholipid condition without CHS.

Since a substantial portion of protomers (<32.6%) were in the PLN conformation in Cx43-M257 GJIChs surrounded by LMNG and CHS (condition 2), neither LMNG nor CHS were strong inhibitors of the NTH−TMD interaction. Given that approximately 41% of PLN protomers were included in Cx43-M257 GJIChs under conditions 8, POPE likely had similar or lower inhibitory effects on the NTH−TMD interaction compared with LMNG/CHS. However, the combination of POPE and CHS may have a higher inhibitory effect than LMNG/CHS because Cx43-M257 GJIChs surrounded by POPE and CHS (condition 7) contain no detectable PLNs.

## The structural determination of the four different conformations

To determine a high-resolution structure in the PLN conformation from dataset 2 containing less than 32.6% of the PLN protomers, we conducted a hemichannel-focused 3D classification to sort the hemichannel sub-particles mainly containing 4–6 PLN protomers. This classification produced one hemichannel sub-structure class (class 6, 7.3%) in the full PLN conformation (Supplementary Fig. 3c). We traced back and gathered the original GJICh particles containing the hemichannel sub-particles in class 6 and performed a 3D reconstruction with C6 symmetry (Supplementary Fig. 3c). This resulted in the GJICh structure comprising full GCN and PLN hemichannels at a high resolution of 3.6 Å, which we used to build an atomic model of the structurally heterotypic channel (Fig. 1d, right, and Supplementary Figs. 3c, 4).

A similar approach was used to classify the hemichannel regions of GJICh particles in dataset 8 (Fig. 1e and Supplementary Fig. 3c). We identified four conformationally different hemichannel sub-structure classes, which were more than expected, because only three conformations were identified in the protomer-focused classification. Since the GJICh structures determined based on this classification would have shown an average of relatively homogeneous conformations in one hemichannel and largely heterogeneous conformations in the opposite hemichannel, we determined four hemichannel substructures at 3.7–4.0 Å and used them to build atomic models (Supplementary Fig. 4). Two of these structures, termed GCN and GCN$^{TMli}$, showed similar GCN conformations but substantial differences in the TM helices (Fig. 1e and Supplementary Figs. 3c, 7a). The other two were in the FIN and PLN conformations, as expected (Fig. 1e and

Supplementary Fig. 3c). Since these four structures were reconstructed from the hemichannel regions of the Cx43 GJICh particles, one should be careful that they are not structures of undocked hemichannels in a single membrane. It should also be noted that all structures contain considerable noise map densities from alternative conformations (See Supplementary Discussion for details).

## The PLN conformation and its comparison with the GCN conformation

The PLN conformation of Cx43 shared many structural features with that of Cx46/50. First, the average Cα deviation was approximately 1.2 Å between the Cx43 and Cx46 protomers (Supplementary Fig. 7d). Second, the W4, L7, and L10 residues in NTH were bound to the hydrophobic surface formed by TM1 and TM2 (Supplementary Fig. 7e). The W4 residue appeared to play a key role in stabilizing the PLN conformation by entering a deep pocket formed by two neighboring TM1s. In addition, the loop structures between NTH, TM1, and the loosely packed unstable helix (π-helix) in the middle of TM1 were similar (Supplementary Fig. 7d).

The Cα deviation between the GCN and PLN hemichannel regions showed substantial changes in the cytoplasmic half of TMD (Supplementary Fig. 7f). In particular, the middle part of TM1 (residues 29–35; LFIFRIL sequence) formed an α-helix in the GCN conformation and a π-helix in the PLN conformation (Fig. 5a and Supplementary Fig. 7a). This sequence is one of the most highly conserved regions in the connexin family, suggesting that structural changes in this region may be functionally important in all connexin GJIChs (Supplementary Fig. 7b, c). The α-to-π-helix transition in TM1 caused a rotation of approximately 55° and approximately 10° of bending for the cytoplasmic half (Fig. 5a, b), resulting in marked changes in the interactions of TM1 with neighboring helices. First, F30 dissociated from TM2 and formed new interactions with two NTHs, while R33 and F32 slightly moved but maintained their original interactions (level 1; Fig. 5c, d). Second, as L29 and L26 moved toward the pore, their interactions with TM4 were replaced by new interactions with TM2 and NTH (level 2; Fig. 5c, d). Third, V28, W25, and V24 of TM1, which participated in tight intermolecular interactions with TM2 of the neighboring protomer in the full GCN hemichannel region, moved away from TM2 and interacted intramolecularly with TM4 in the full PLN hemichannel region (level 2; Fig. 5c, d). This change resulted in a small gap between the protomers, and the movement of S27 toward the gap slightly reduced the strong hydrophobicity of this region (level 2; Fig. 5d and Supplementary Movie 2). In particular, W25, exposed to the membrane in the GCN state, moved into the hydrophilic center of TMD, participating in hydrogen bonding with S220 and a cation·π interaction with K162 (level 2; Fig. 5c, d). Fourth, in the GCN state, L11 and Q15 of NTH interacted with F97 and Y98 of TM2 in a neighboring protomer. V24 of TM1 also interacted with Y98 (level 3; Fig. 5c). F97 and L11 closely interacted with CHS2 and POPE/LMNG between neighboring NTHs. However, in the PLN state, these interactions disappeared, and an opening toward the membrane was created between the protomers (level 3; Fig. 5d).

In addition to TM1, other TM helices underwent substantial structural changes during the GCN-to-PLN transition. The cytosolic halves of TM2, 3, and 4 rotated slightly around TM1, and the loop connecting NTH and TM1 (residues 15–20) moved approximately 3 Å toward the pore inside (Supplementary Fig. 8a–c and Supplementary Movie 3). Consequently, the NTH−TM1 loop mainly interacted with TM3 and TM4 in the GCN conformation and with TM2 in the PLN conformation (Supplementary Fig. 8b, c).

## The GCN$^{TMli}$ conformation

One of the two GCN hemichannel sub-structures (condition 8), which accounted for 7.3%, was nearly identical to the structure of the full GCN GJICh in POPE/CHS (conditions 6 and 7) (Fig. 1d, e). The superposition of

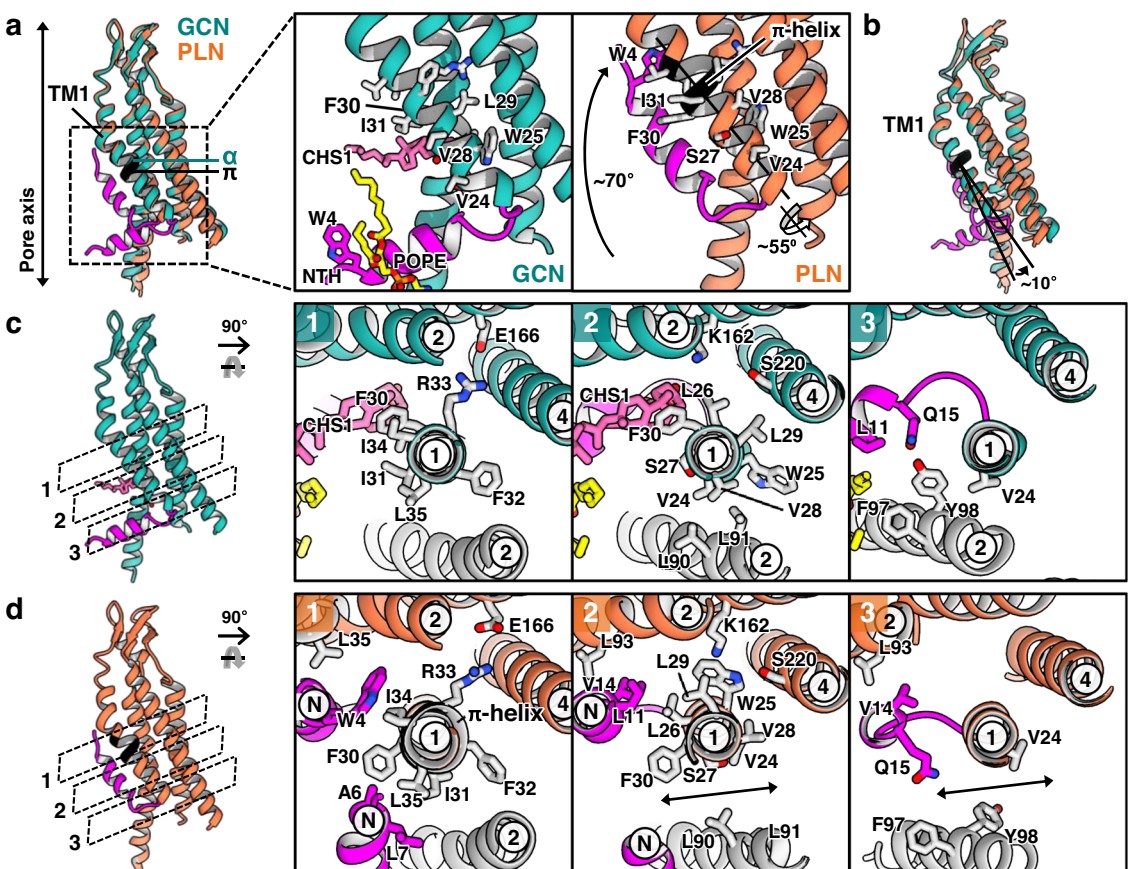

**Fig. 5 | Conformational changes of NTH and the α-to-π helix transition in the TM1 during the conversion of GCN to PLN. a** Structural alignment of Cx43-M257 protomers in GCN and PLN conformations (left). The α- and π-helices of protomers in the GCN and PLN conformations are labeled, respectively. Close-up view of the boxed region (right). Movements of NTH and rotations of TM1 are indicated by one-way arrows. Residues in TM1 and W4 are drawn as sticks and labeled. The lipids in the GCN protomers from these three structures showed an average Cα deviation of 0.6 Å. By contrast, the protomers in the other GCN hemi-channel sub-structure (referred to as GCN^TMIi, accounting for 20.4%) adopted π-helix instead of α-helix in the middle of TM1 and showed an incomplete rotation of the cytoplasmic part of TM1 (Fig. 1e and Supplementary Figs. 7a, 9a, b). This consequently caused the conformations of other TM helices to differ from those of the GCN and PLN protomers.

The map density of a POPE molecule between two neighboring NTHs was clearly observed in the GCN and GCN^TMIi hemichannel sub-structures. However, that of CHS1 was unclear in the GCN^TMIi structure, and CHS2 was barely detectable in either structure (Supplementary Fig. 9c). In addition, the side-chain conformation of F30 in TM1, which participates in the stacking interaction with the sterol ring of CHS1 in the GCN structure, was largely changed in the GCN^TMIi structure (Supplementary Fig. 9a). These observations verified that the protein sample contained only a small amount of CHS, indicating that the majority of GCN protomers in this structure have no bound CHS. Therefore, we concluded that the GCN conformation can be formed in phospholipids without CHS.

**The FIN conformation**
The map density of the FIN hemichannel sub-structure contained substantial noise from different NTH conformations as the FIN protomer class was the minor class (22%; Supplementary Fig. 3a). Nevertheless, the map densities of the TM helices were homogeneous and similar to those of the PLN hemichannel sub-structure, including the π-helix in the

TM1 are also drawn and labeled. **b** Structures in (**a**) are rotated by 30° to show the movement of TM1 in the PLN protomer during the α-to-π-helix transition. **c, d** GCN and PLN protomers were cross-sectioned at three levels (gray dotted lines) along the helical axis of TM1 and individually viewed from the extracellular side. TM2 in the neighboring protomer is colored gray. Circled numbers and 'N' indicate the TM helix numbers and NTH, respectively.

middle of TM1 and large gap between two neighboring protomers (the membrane opening; Fig. 6a and Supplementary Figs. 7a, 9d, 10a). When the final model of the FIN protomer was superimposed on that of the PLN protomer, we found only a slight movement of TM helices in their cytoplasmic regions and a 50° kink of NTH toward the cytoplasm (Figs. 1e and 6b). Another superposition with the GCN protomer showed a 20° kink toward the extracellular region (Figs. 1e, 6b). This kink, together with the movement of TM1 and TM2 during the GCN-to-FIN transition, half-blocked the CHS1-binding site and formed a hydrophobic path from the site to the membrane opening (Fig. 6c and Supplementary Fig. 9d). This path was also near the POPE-binding site between two neighboring NTHs (Fig. 6c). Therefore, it is possible that during the GCN-to-FIN transition, the bound CHS and POPE molecules inside the channel diffused out of the channel via a hydrophobic path. However, this has yet to be demonstrated experimentally.

Although the NTH densities in the FIN hemichannel sub-structure were strong, they were not separated from each other. In addition, four N-terminal residues (residues 2–5) were not observed (Fig. 1e). These results indicate that the FIN conformation is relatively flexible compared with the PLN and GCN conformations. Because the six N-termini in the FIN hemichannel sub-structure are too close to each other to avoid steric hindrance, the full FIN structure should not be stable. The distribution of FIN protomers also showed that few hemichannel sub-particles contained five and six FIN protomers (Fig. 4b). Therefore, we concluded that the full FIN structure is artificial, and only its protomer structure should be used.

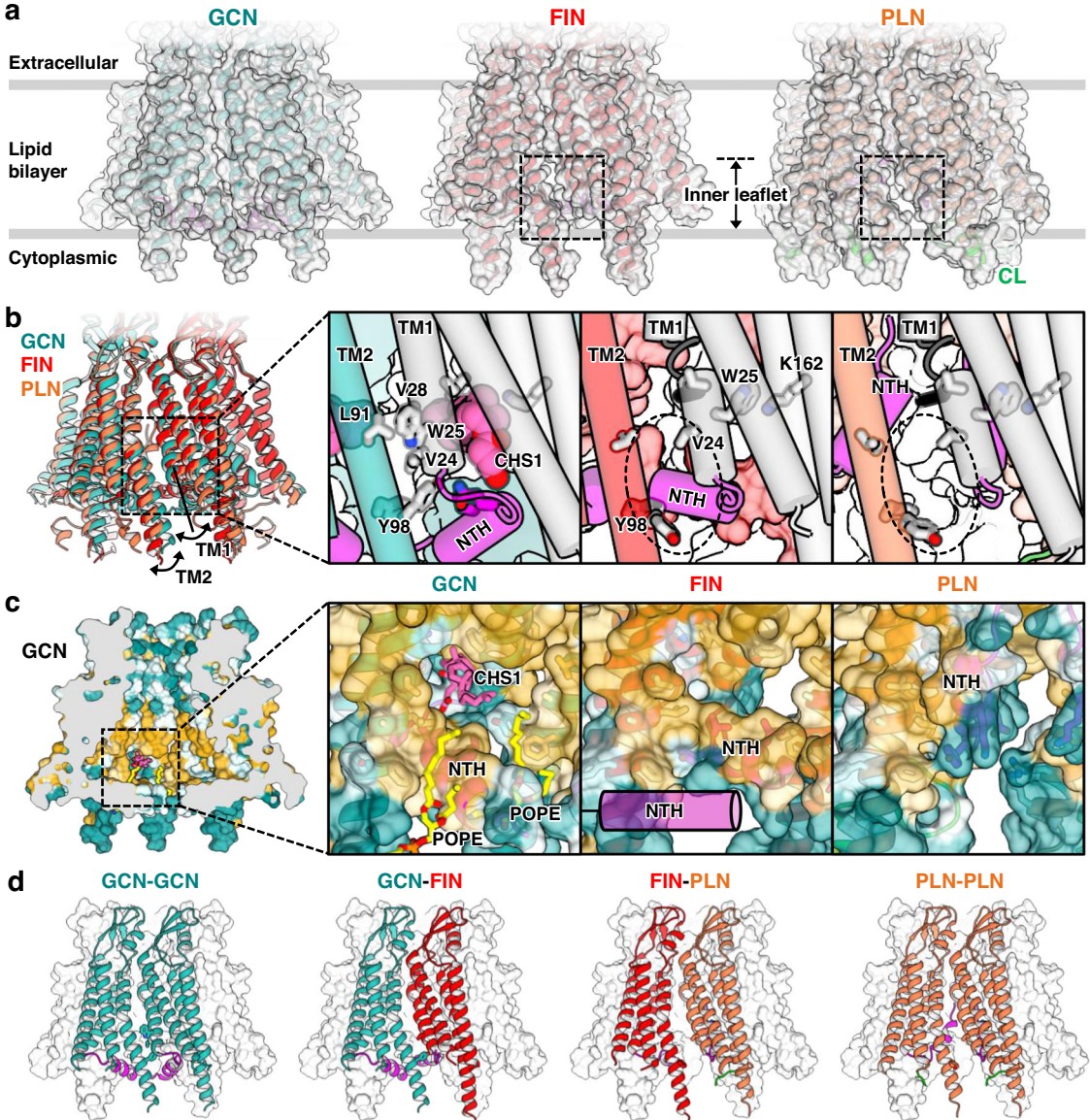

**Fig. 6 | Membrane openings in the FIN and PLN hemichannel sub-structures. a** Surface representations of the full GCN, FIN, and PLN hemichannel sub-structures. The membrane openings are shown in the FIN and PLN hemichannel sub-structures. **b** Structural alignment of the three hemichannel sub-structures in (**a**) (left) and close-up view of the boxed region for each hemichannel sub-structure (right). Helices and residues related to the membrane opening are drawn as cylinders and sticks, respectively. The membrane openings are indicated by the dotted circles. **c** Sliced view of Cx43 hemichannel sub-structures in (**a**), viewed from the membrane (left), and close-up view of the lipid-binding pocket in each hemichannel sub-structure (right). A neighboring NTH in the full FIN hemichannel sub-structure is drawn as a cylinder to show the reduced size of a hole between neighboring NTHs. Surface hydrophobicity is shown as in Fig. 3b. Hydrophobic and hydrophilic residues are colored yellow and green, respectively. **d** Membrane openings formed between conformationally different protomers. Only two consecutive protomers are shown for clarity. The membrane opening in the FIN–PLN pair was slightly larger than that in the PLN–PLN pair.

The FIN conformation was not identified under the tested conditions other than condition 8. The conformation seems to readily change to GCN in high CHS content, but we cannot exclude the possibility that it is specific to the M257 mutation until it is confirmed in Cx43-WT structures.

**Pore sizes in various compositions of PLN and GCN protomers**
The constriction diameter in the pore pathway in the full PLN conformation was approximately 10 Å (Fig. 7a, b), which was larger than that in the full GCN conformation (approximately 5 Å). To determine whether a hemichannel region with conformationally varying protomers would have a larger pore size, we built artificial hemichannel models with all of the possible combinations of PLN and GCN protomers by superposing the protein regions spanning ECLs and the

extracellular halves of TMDs, whose structures are almost identical between GCN and PLN protomers (Fig. 5a). We confirmed that there was almost no steric hindrance at the interface between the GCN and PLN protomers (Fig. 4d). Because we did not include lipid molecules inside the channel pore, the analysis with these models assumed that the GCN conformation in each protomer was temporarily possible without lipids in a channel, or that lipids inside the pore were mobile and, thus, do not substantially affect the pore size. The lipids would be considerably mobile and flexible in the conformations with more than two PLN protomers, where the pore surface is moderately or highly hydrophilic.

We analyzed the solvent-accessible pore diameters of these hemichannel models and found that the constriction diameters of two hemichannels with three and four consecutive PLN protomers were

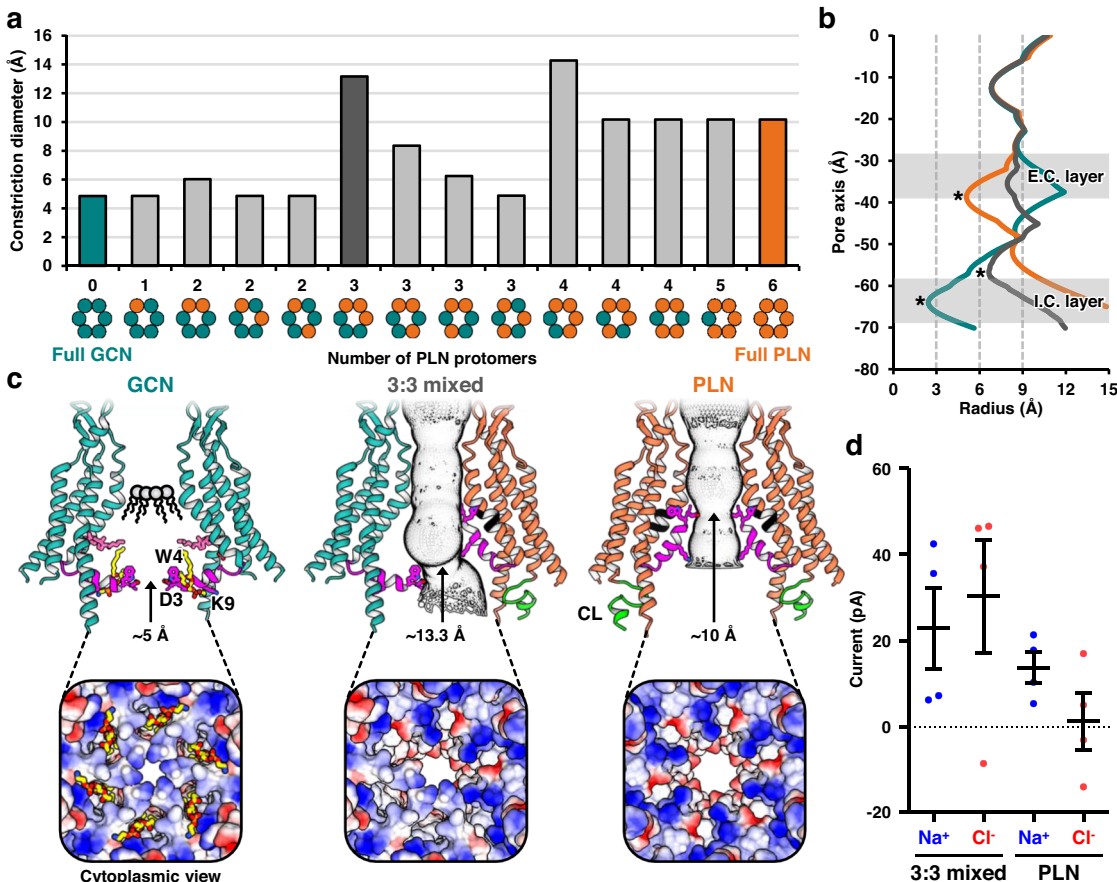

**Fig. 7 | Changes in the pore properties during the GCN-to-PLN conversion of individual protomers. a** Constriction diameters of various hemichannel regions with possible arrangements of the PLN protomers. **b** Comparisons among the solvent-accessible pore radii changing along the pore pathways of GCN, 3:3 mixed, and PLN hemichannel regions are depicted. Pore constriction sites are indicated by asterisks. Extracellular layer (E.C. layer) and intracellular layer (I.C. layer) of pore-occluding lipids are indicated. **c** Comparisons among GCN, 3:3 mixed, and PLN GJIChs are depicted with pore pathways (top). For clarity, only two protomers facing each other are shown. The NTHs are colored magenta. The π-helix in the middle of TM1 is black. Surface electrostatic potentials of the pore region viewed from the cytoplasmic side (bottom). The surface electrostatic potentials are calculated and colored using Coulombic Surface Coloring in Chimera. The displayed potentials range from −10 (red) to +10 (blue) $kT$/e. CHS1 and POPE are drawn as sticks. The constriction diameters are labeled. **d** Ionic currents and selectivity of 3:3 mixed and full PLN GJIChs in the MD simulation. Data are presented as mean ± SE ($n = 4$). Source data are provided as a Source Data file.

13.2 and 14.3 Å, respectively (Fig. 7a). This observation suggests that large molecules may pass through the channel more efficiently in the mixed conformation than in the full PLN conformation.

**Electrostatic surface potential and ion selectivity of Cx43 GJICh**
In the electrostatic surface potential representations of Cx43 GJICh, the cytoplasmic surface of the full GCN hemichannel sub-structure had an overall positive charge, whereas that of the full PLN hemichannel sub-structure showed a locally negative surface in the pore pathway (Fig. 7c and Supplementary Fig. 11a). This remarkable change occurs because NTHs not only bend, but also rotate during the GCN-to-PLN transition. During the rotation of NTH, basic residues (K9 and K13) exposed on the cytoplasmic surface in the GCN conformation were partially buried at the interface between the NTHs in the PLN conformation, and acidic residues (D3 and D12) were exposed on the cytoplasmic surface of the funnel-like pores (Supplementary Fig. 11b). This suggests that the ion transmission efficiency and selectivity may be largely affected by conformational changes in individual NTHs in the channel.

To test this hypothesis, we chose two conformationally different Cx43 GJIChs, the full PLN GJICh and a GJICh with three consecutive PLN and GCN protomers in each hemichannel (referred to as 3:3 mixed GJICh). We manually reconstructed the CL of each Cx43 chain in a random configuration to maintain the connectivity between TM2 and TM3. We added two POPC molecules to block two holes between the three consecutive GCNs in the GJICh model and did not include lipid molecules at the extracellular layer because of their potential mobility and flexibility in a considerably hydrophilic pore.

We conducted MD simulations to determine the position-dependent local concentration and current of Na$^+$ and Cl$^-$ at a 200-mV transmembrane potential[33] (see the gray heatmap for the local concentration and streamlines for the current study; Supplementary Fig. 11c, d). In the full PLN GJICh, we found a large Cl$^-$ reservoir at the channel entrance (z = approximately 6 nm) with a local Cl$^-$ concentration > 0.8 M, whereas a small Na$^+$ reservoir (z = approximately 2.5 nm, > 0.4 M) was formed at the Na$^+$-leaving hemichannel. This resulted in the Na$^+$ current (13.6 ± 3.6 pA) being much higher than Cl$^-$ current (1.2 ± 6.5 pA; Fig. 7d and Supplementary Fig. 11c, d). The calculated conductance was 74 pS. However, in the 3:3 mixed GJICh, a shallow Cl$^-$ reservoir was formed at the entrance (z = approximately 6 nm) with a local Cl$^-$ concentration > 0.4 M. The Cl$^-$ fluxes coming in and out the reservoir were large, explaining a high Cl$^-$ current of 30.3 ± 13.1 pA. A deeper Na$^+$ reservoir (z = approximately 3 nm, >0.8 M) at the Na$^+$-leaving hemichannel and a larger Na$^+$ flux out the reservoir resulted in a higher Na$^+$ current of 22.8 ± 9.4 pA than that in the full PLN GJICh (Fig. 7d and Supplementary Fig. 11c, d). The calculated conductance was 265.5 pS. In all simulations, flexible CLs appeared not to prevent ion fluxes coming into or out of the pore, and thus the

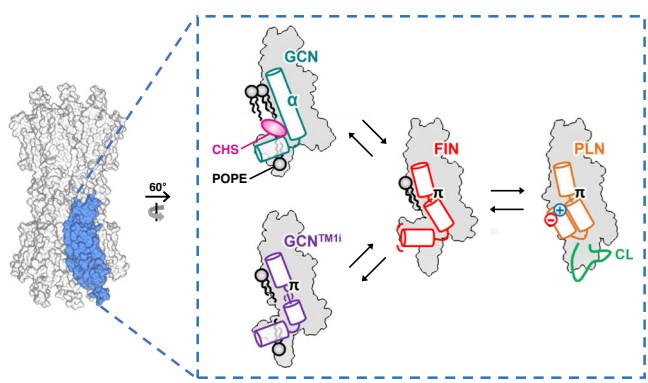

**Fig. 8 | Schematic diagram of a dynamic conformational equilibrium of Cx43 GJICh in POPE-nanodiscs with a low CHS content.** The structural change in only one protomer is depicted. POPE and CHS are colored in black and pink, respectively, and labeled. In the middle of TM1, α- and π-helices are indicated as α and π, respectively. The green line in the PLN protomer indicates partial stabilization of CL.

conductance was likely determined by the pore structures and surface properties created by TMDs and NTHs. However, since CL is structurally unknown and CTD is not included in the Cx43 models used for the simulations, these data need to be carefully interpreted considering the absence of the gating regulation by CL or CTD in the simulations.

## Discussion

Connexin GJIChs are regulated by many factors and their proper regulations are crucial for numerous biological processes. However, their mechanisms of action and regulation are not clearly understood due to insufficient structural information. To understand the structural changes of GJICh related to those mechanisms, we performed the cryo-EM studies on Cx43 GJICh under various conditions. We determined 12 cryo-EM structures of the channel and found four different conformations of Cx43 (GCN, GCN$^{TMIi}$, FIN, and PLN). Conformational variation analyses elucidated dynamic changes in individual protomers in a single GJICh, which consequently modified the size and surface properties of the pore pathway. These findings suggest that the channel activity may be regulated by conformational changes in individual protomers (Fig. 8). The conformational change in each protomer from GCN to PLN likely includes three steps: first, the α-to-π-helix transition in the middle of TM1 in the GCN conformation; second, the rotation of the cytoplasmic part of TM1 and formation of the FIN conformation; and third, the release of lipids from the pore interior and formation of the PLN conformation (Fig. 8).

Cx43 GJICh exhibited various compositions of four protomer conformations in phospholipid nanodiscs (condition 8; Fig. 1c, e). The conformational diversity of the channel captured by cryo-EM indicated that this channel is in a dynamic conformational equilibrium of individual protomers. The relative proportions of conformationally different protomers were substantially changed by different detergent/lipid environments, partial deletion of CTDs, and pH changes (Supplementary Fig. 3a). This suggests that the NTH−TMD interaction for channel opening is inhibited or enhanced to different degrees by various factors and excludes the possibility that the structural variation is motionlessly maintained by the strong binding of any substance in the protein sample. These findings would be valuable information for the experimental design and data interpretation to understand the action or regulation mechanisms of GJIChs. However, the involvement of lipids in the conformational change of Cx43 GJICh needs to be carefully interpreted. There is currently no evidence that lipids play a physiological role in the gating of Cx43 GJICh. In addition, we cannot exclude the possibility that the interactions of the pore pathway with lipids or detergents were artificially introduced and maintained during

sample manipulation. The observed conformational changes of the channel might be related to the intercellular transfer of soluble molecules with substantial hydrophobicity. Further studies are needed to clarify whether membrane lipids can directly regulate the channel activity.

The independent conformational changes of individual protomers in Cx43 GJICh resulted in various pore sizes (constriction diameters of 4.9–14.3 Å) and, thus, were relevant to the channel gating. However, it is unclear whether Cx43 GJIChs in cell membranes are in dynamic equilibrium with four different protomer structures. Although the channels have only 41% PLN protomers in the POPE environment (condition 8), we cannot exclude the possibility that an abundant cellular component may completely push the equilibrium to the full PLN state for the channel opening. In addition, full GCN GJICh occluded by lipids might not be a physiologically relevant state or the only closed state. Therefore, it is still unclear which structures represent the closed, residual, and/or open states induced by known regulatory factors such as Vj, pH, and Ca$^{2+}$/calmodulin. However, an electrophysiological study on a chimeric connexin hemichannel shows a possibility that the presented structures in this study may be relevant to Vj-gating. The study has shown a series of gating transitions in Vj gating, suggesting that this hemichannel underwent conformational changes in individual protomers rather than a concerted change of all six protomers[34]. Another study showed that a mutant Cx43 GJICh containing D12S and K13G was not closed by a high junctional voltage[22]. D12 and K13 closely interact with TM2 to stabilize the GCN conformation, but not the PLN conformation. This suggests that the GCN conformation may represent the closed state in Vj-dependent gating. Further structural and functional studies are needed to address many questions regarding various gating regulations. The advance of cryo-electron tomography would be required to directly measure or show small changes in NTHs for GJICh in a cell membrane. Although mutational studies can be performed using currently available techniques, it seems difficult to identify proper mutations that shift the conformational equilibrium without removing channel activity. An alternative method would be to find at least two molecules that shift the conformational equilibrium to PLN and GCN using cryo-EM experiments, respectively, and then measure the effects of the molecules on ion transmission through the channels using electrophysiology experiments.

Although we could not clearly indicate the structure of Cx43 GJICh that represented the fully open state, our structural analysis and MD simulation suggest that the maximally open channel may not be the full PLN GJICh, but the channel in the dynamic conformational equilibrium. Compared with the full PLN GJICh, a GJICh containing one or two GCN protomers in each hemichannel had a similar or bigger pore (Fig. 7a), and the 3:3 mixed GJICh showed more than three times larger conductance in the MD simulation. While sulforhodamine B[35] with the smallest diameter of approximately 12 Å may be too big to pass through the channel in the full PLN state, it may be able to pass through the channel in the dynamic conformational equilibrium because pores with a diameter >13 Å can be frequently formed during conformational changes between GCN and PLN protomers. MD simulations additionally showed little ion selectivity for the 3:3 mixed GJICh and strong cation selectivity for the full PLN GJICh (Fig. 7d). This also supports our hypothesis because previous electrophysiological studies on rat Cx43 GJICh have mostly shown no ion selectivity in the maximally open state[36]. Furthermore, the measured conductances (105–120 pS) of rat Cx43 GJICh in the fully open state[24,27,36] are between the calculated conductances of 3:3 mixed and full PLN GJIChs (265.5 pS and 74 pS) in our MD simulation, suggesting that the open state may be in a dynamic conformational equilibrium between the two conformations. One exceptional study with various ionic environments showed strong cation selectivity and a conductance of 79 pS in the NaCl environment[37], suggesting that the

channels in this study may have been in the full PLN state under cell-specific conditions.

Supplementary Discussion includes the results and/or discussions for a potential role of cholesterols in the channel gating, considerable noise in four hemichannel sub-structures, evolutionary conservation of the structural changes in TM1 and NTH, membrane opening, structural comparison between Cx43 and Cx46/50 in the PLN conformation, conformational changes in the CL during the GCN-to-PLN transition, structural comparison between Cx43 and Cx31.3 in the GCN conformation, detailed structure of H-type interconnexon docking interactions, and mapping disease-causing mutations on the full GCN and PLN GJICh structures.

## Methods

### Plasmid construction

A synthetic gene fragment encoding the full-length human Cx43 (*GJA1*) (UniProt ID # P17302) was purchased from Integrated DNA Technologies and inserted into pX vector as previously reported[38]. The human Cx43-WT proteins were expressed as fusion constructs with a human rhinovirus (HRV) 3C cleavage site, an enhanced yellow fluorescence protein (eYFP) tag, a 10×His-tag and a rho-1D4 epitope tag (8 amino acid sequence of TETSQVAPA) at its C-terminus. To make a Cx43-M257 expression plasmid, the PCR-amplified gene fragment encoding 1-257 amino acids of human Cx43 was inserted in pX vector.

### Protein expression and purification of human Cx43-WT

Cx43-WT expression plasmids were amplified in *Escherichia coli* DH5α strains, purified, mixed with linear polyethylenimine (PEI, Polysciences, Inc.), and transfected into human embryonic kidney 293E (HEK293E) cells (ATCC, CRL-10852)[9,38]. When the HEK293E suspension cells (2 L) were grown at 37 °C to a density of ~0.6 × 10^6 cells/ml in Ca^2+^-free Dulbecco's modified Eagle's medium (DMEM) supplemented with 5% fetal bovine serum, the mixtures of DNA and PEI were added to the cells. Dimethyl sulfoxide (Amresco) was added to the final concentration of 1%, and temperature was lowered to 33 °C immediately after transfection. Tryptone (Amresco) was added 48 h after transfection to the final concentration of 0.5%. The transfected cells were harvested 96 h after transfection.

All purification steps were carried out at 4 °C unless indicated otherwise. The harvested cells were resuspended in buffer A [20 mM CAPS (pH 10.5), 250 mM KCl, and 2 mM β-mercaptoethanol] supplemented with 10% glycerol, 2 µg/ml *Staphylococcus aureus* nuclease (SA nuclease), 5 mM CaCl$_2$, 1 mM phenylmethylsulfonyl fluoride (PMSF), and one tablet of EDTA-free Pierce protease inhibitor tablets (Thermo Fisher Scientific, catalog no. 88666). The resuspended cells were lysed using a Dounce homogenizer (Bellco) with a tight (B) pestle (25–30 strokes), and the membrane fraction was isolated by high-speed centrifugation at 42,600 g for 1 h. The membrane pellets were resuspended using a WiseTis homogenizer (Daihan Scientific Co., Ltd.) in 50 ml buffer A supplemented with 2% glycerol, 1 mM PMSF, 2 mM EDTA, 2 mM EGTA, one tablet of EDTA-free Pierce protease inhibitor tablets, and 0.5/0.05% (w/v) LMNG/CHS (Anatrace, catalog no. NG310-CH210). After incubation for 1 h with slow rotation, the samples were mixed with 2.5 ml of neutralization buffer containing 1 M Tris (pH 7.5) to lower the pH of samples to ~8.0 and centrifuged at 42,600 g for 1 h. The supernatant was mixed with adipic acid dihydrazide-agarose resin (Sigma, catalog no. A0802) conjugated with rho-1D4 antibody (University of British Columbia) in an open column (Bio-Rad) and incubated for 1 h with gentle rotation. The resins were settled down in the column and washed twice with 10 column volumes (CVs) of buffer B [20 mM Tris (pH 8.0), 250 mM KCl, 2 mM β-mercaptoethanol, 2% glycerol, and 0.004/0.0004% (w/v) LMNG/CHS], and once with 10 CVs of buffer C [20 mM Tris (pH 8.0), 250 mM KCl, 2 mM β-mercaptoethanol, and 0.004/0.0004% (w/v) LMNG/CHS]. The bound proteins were incubated for 2 h with the addition of excess HRV 3C protease

(~0.25 mg) to cleave off the C-terminal eYFP-rho-1D4 tag from Cx43-WT proteins, and the proteins were eluted from the resin. To remove possible post-translational modifications (PTMs) on Cx43, the eluted proteins were incubated for 2 h with lambda protein phosphatase (λ-PPase) and universal deubiquitinase DUB (M48) at a 15:1 (w/w) ratio of Cx43 to the enzymes. For the activation of λ-PPase, 1 mM MnCl$_2$ was added to the reaction buffer, and the reaction was stopped by adding 5 mM EDTA after 2 h. Next, the proteins were concentrated to ~3 mg/ml using an Amicon Ultra centrifugal filter (molecular weight cutoff, 100 kDa), filtered with a 0.22 µm filter, and then loaded on a Superose 6 Increase 10/300 column (GE Healthcare) equilibrated with buffer C. Peak fractions were pooled, concentrated to ~3.2 mg/ml, flash-frozen in liquid nitrogen, and stored at −80 °C for nanodisc reconstitution and EM grid preparation. Protein purity and quality were assessed by SDS-polyacrylamide gel electrophoresis (SDS-PAGE) and negative-stain EM. The protein sample treated with λ-PPase and M48 during purification showed a homogeneity much higher than that of the untreated sample on SDS-PAGE (Supplementary Fig. 1m).

Purification of Cx43-WT sample in GDN (Anatrace, catalog no. GDN101) was also carried out as described above, with slight modifications. Briefly, GDN instead of LMNG/CHS was used with a concentration of 1% and 0.01% (w/v) during extraction and gel filtration steps, respectively. Protein samples were concentrated to ~3.1 mg/ml, flash-frozen in liquid nitrogen, and stored at −80 °C for EM grid preparation.

### Protein expression and purification of Cx43-M257

Cx43-M257 was expressed in HEK293E cells in the same method as in the Cx43-WT expression. Protein purification was carried out at 4 °C. The transfected cells were lysed using a Dounce homogenizer in buffer D [20 mM Tris (pH 8.0), 250 mM NaCl, 10% glycerol, 0.1 mM PMSF, one tablet of EDTA-free Pierce protease inhibitor tablets, RNase A, and SA nuclease]. The lysate was centrifuged at 35,000 g for 1 h, and the pellets were washed in buffer E [20 mM (pH 8.0), 1 M NaCl, 1 mM EDTA, 2% glycerol, and protease inhibitor cocktail]. Membrane proteins were extracted in buffer F [20 mM CAPS (pH 10.5), 150 mM KCl, 2% glycerol, 2 mM β-mercaptoethanol, 1 mM EDTA, 0.5/0.05% (w/v) LMNG/CHS, and protease inhibitor cocktail] for 1 h with gentle rotation and then centrifuged at 20,000 g for 1 h. The supernatant was mixed with agarose resin conjugated with rho-1D4 antibody at 4 °C for 2 h and washed three times with 10 CVs of buffer G [20 mM Tris (pH 8.0), 150 mM KCl, and 2 mM β-mercaptoethanol] supplemented with 2% glycerol, and 0.005/0.0005% (w/v) LMNG/CHS. Bound proteins were eluted by adding a 1:20 (w/w) of HRV 3C protease and overnight incubation. The eluted protein was concentrated and loaded on a Superose 6 Increase 10/300 column equilibrated with buffer G supplemented with 0.005/0.0005% (w/v) LMNG/CHS. Peak fractions were pooled, concentrated to ~3 mg/ml, flash-frozen in liquid nitrogen, and stored at −80 °C. The treatment with λ-PPase and M48 was not applied for Cx43-M257, because most PTM sites are located on Cx43 CTD.

### Expression and purification of membrane scaffold proteins

The membrane scaffold proteins (MSP1E3D1 and MSP1E1) were expressed and purified as previously described[39], with slight modifications. The pET28a plasmids containing MSP1E3D1 and MSP1E1 gene were obtained from Addgene (plasmid #20066 and #20062, respectively). For expression of MSP1E3D1 and MSP1E1, *E. coli* BL21 (DE3) competent cells were transformed with pET28a-MSP1E3D1 and pET28a-MSP1E1, respectively, and grown on SOC plates containing 30 µg/mL kanamycin. A single colony was picked and cultured in terrific broth supplemented with 30 µg/mL kanamycin at 37 °C. When the cell density reached an OD$_{600}$ of 1.2-1.4, protein expression was induced by 1 mM IPTG for 3 h at 37 °C. The cells were harvested by centrifugation at 8000 g for 30 min. The cell pellet was resuspended in buffer H [20 mM Tris (pH 8.0), and 250 mM NaCl] supplemented with

10% glycerol, 1% (w/v) Triton X-100, and 1 mM PMSF. The cells were lysed by sonication and centrifuged at 40,000 g for 1 h. The supernatant was loaded onto a nickel-nitrilotriacetic acid (Ni-NTA) resin equilibrated with buffer I [20 mM Tris (pH 8.0), 250 mM NaCl, and 1% (w/v) Triton X-100]. The Ni-NTA resin was washed by the following buffers (5 CV each): buffer I, buffer I supplemented with 20 mM imidazole, buffer H supplemented with 40 mM imidazole. The bound proteins were eluted by buffer H supplemented with 300 mM imidazole. To remove imidazole, the eluted proteins were loaded onto a HiPrep 26/10 desalting column (GE Healthcare) equilibrated with buffer H. The eluted MSP1E3D1 and MSP1E1 samples were supplemented with 1 mM EDTA, concentrated to ~3.6 mg/ml and ~4.6 mg/ml, respectively, flash-frozen in liquid nitrogen, and stored at −80 °C.

### Reconstitution of Cx43-WT in lipid nanodiscs

Purified Cx43-WT was reconstituted into lipid nanodiscs as previously described[39], with some modifications. To make Cx43-WT samples in nanodiscs containing soybean lipids and POPE/CHS (POPE:CHS = 85:15, w/w) lipid mixture, soybean polar lipid extract powder (Avanti Polar Lipids) and POPE powder (Avanti Polar Lipids) with CHS, respectively, were dissolved in chloroform, aliquoted, and dried completely to be a thin layer using argon stream. Just before reconstitution, the lipid film was solubilized in 5/0.5% (w/v) LMNG/CHS, and incubated at 60 °C for 30 min to make a clear lipid stock solution at a concentration of ~10 mg/ml.

All the reconstitution steps were carried out at 4 °C. Cx43-WT samples (~3.2 mg/ml, ~700 μl) were mixed with the lipid stocks at the 1:66 molar ratio of Cx43-WT to lipid, and incubated for 20 min. Next, MSP1E3D1 was added to the Cx43-lipid mixtures at ~1:0.3 molar ratio of Cx43-WT to MSP1E3D1 and was incubated for 40 min. Bio-beads SM2 absorbent (120 mg per 1 ml mixture, Bio-Rad) pre-washed with buffer G were added to the Cx43-lipid-MSP mixtures to remove detergents and were incubated overnight with slow rotation. Bio-beads were removed by centrifugation and fresh Bio-beads were added and incubated for 4 h to remove residual detergents. After Bio-beads were removed, the reconstituted mixtures were concentrated to reduce the sample volume up to 500 μl, filtered with a 0.22 μm filter, and loaded on a Superose 6 Increase 10/300 column equilibrated with buffer G. Peak fractions containing Cx43 samples in nanodiscs (consisting of soybean lipids) and in nanodiscs (POPE/CHS) were pooled, and concentrated to ~2.2 mg/ml and ~4.5 mg/ml, respectively. The reconstitution quality was assessed by SDS-PAGE and negative-stain EM.

### Reconstitution of Cx43-M257 in lipid nanodiscs

Purified Cx43-M257 samples were reconstituted into lipid nanodiscs as described above, with some modifications. The POPE powder was dissolved with or without CHS in chloroform, dried completely using argon stream to make lipid films for POPE/CHS (POPE:CHS = 85:15, w/w) and POPE, respectively, and stored at −80 °C prior to reconstruction. The lipid films for POPE/CHS and POPE were solubilized in 5/0.5% (w/v) LMNG/CHS and 5% (w/v) LMNG, respectively, and incubated at 60 °C for 30 min to make lipid stock solutions at a concentration of ~10 mg/ml.

All the reconstitution steps were performed at 4 °C. Cx43-M257 samples (~1.1 mg/ml, ~750 μl) were mixed with the lipid stocks at the 1:90 molar ratio of Cx43-M257 to lipid, and incubated for 20 min. After MSP1E1 was added to the Cx43-lipid mixtures at 1:0.3 molar ratio of Cx43-M257 to MSP1E1, the Cx43-lipid-MSP mixtures were incubated for 45 min. Bio-beads SM2 absorbent pre-washed with buffer G were added to the mixtures and incubated for 8 h. After Bio-beads were removed by centrifugation, fresh Bio-beads were added and incubated overnight. The reconstituted mixtures were loaded onto a Superose 6 Increase 10/300 column equilibrated with buffer G. Peak fractions containing Cx43-M257 samples in nanodiscs (POPE/CHS) and in

nanodiscs (POPE) were concentrated to ~4.5 mg/ml and ~5.7 mg/ml, respectively. The reconstitution quality was assessed by SDS-PAGE.

### Cryo-EM grid preparation and data collection

The purified Cx43 GJICh samples in detergents or in lipid nanodiscs are in buffer at pH 8.0. For the grid preparation of Cx43-WT and Cx43-M257 samples in LMNG/CHS at pH 8.0, 3 μl protein samples (1.4 mg/ml for Cx43-WT and 2.3 mg/ml for Cx43-M257) were applied onto glow-discharged (15 mA current; negative charge; 60 s) holey carbon grids (Quantifoil R1.2/1.3 Cu 200 mesh, SPI). The grids were blotted for 6.5 s using a Vitrobot Mark IV (Thermo Fisher Scientific, USA) at 4 °C with 100% humidity and vitrified by plunging into the liquid ethane cooled by liquid nitrogen. For preparation of Cx43-WT samples in LMNG/CHS at pH 6.9, a buffer containing 500 mM HEPES (pH 6.8), 150 mM KCl, 2 mM β-mercaptoethanol, and 0.004/0.0004% (w/v) LMNG/CHS was added to the Cx43-WT samples in LMNG/CHS at pH 8.0 at a ratio of 1:3 (v/v), just before vitrification. After three microliters of protein samples (0.7 mg/ml) were loaded onto holey carbon grids (Quantifoil R1.2/1.3 Cu 200 mesh), the grids were blotted and vitrified in Vitrobot. The conditions of glow-discharging and blotting were the same as mentioned above. For the grid preparation of Cx43-WT samples in GDN at pH 8.0, 3 μl protein samples (3.0 mg/ml) were applied onto glow-discharged (15 mA current; negative charge; 60 s) holey carbon grids (Quantifoil R2/2 Cu 200 mesh). The grids were blotted for 6.0 s and vitrified in Vitrobot. For the grid preparation of Cx43-WT samples in nanodiscs at pH 8.0, 3 μl protein samples (1.7 mg/ml and 4.3 mg/ml for nanodisc samples (soybean lipids and POPE/CHS, respectively)) were applied onto glow-discharged (15 mA current; positive charge; 60 s) holey carbon grids (Quantifoil R1.2/1.3 Cu 200 mesh). For Cx43-WT samples in nanodiscs (soybean lipids), glycerol was added to the protein samples at a final concentration of 1% (v/v) to make protein particles disperse evenly. The grids for Cx43-WT samples in nanodiscs (soybean lipids) and in nanodiscs (POPE/CHS) were blotted for 7.0 s and 6.5 s, respectively, and vitrified in Vitrobot. For the grid preparation of Cx43-M257 samples in nanodiscs at pH 8.0, 3 μl protein samples (3.5 mg/ml and 5.7 mg/ml for nanodisc samples (POPE/CHS and POPE, respectively)) were applied onto glow-discharged (15 mA current; negative charge; 60 s) holey carbon grids (Quantifoil R1.2/1.3 Cu 200 mesh). The grids were blotted for 7.0 s and vitrified in Vitrobot.

Three datasets of Cx43 samples in LMNG/CHS (Cx43-WT at pH 8.0 and pH 6.9, and Cx43-M257 at pH 8.0) and two datasets of Cx43-WT samples in nanodiscs (soybean lipids) and in nanodiscs (POPE/CHS) at pH 8.0 were collected at Korea Basic Science Institute (KBSI, Ochang, Korea) using a Titan Krios G2 microscope (Thermo Fisher Scientific, USA) at 300 kV equipped with a Falcon 3EC detector (Thermo Fisher Scientific, USA). Two datasets of Cx43-M257 samples in nanodiscs (POPE/CHS) and in nanodiscs (POPE) at pH 8.0 were collected at Institute for Basic Science (IBS, Daejeon, Korea) using a Titan Krios G4 microscope (Thermo Fisher Scientific, USA) at 300 kV equipped with a K3 detector (Gatan, USA) and BioQuantum energy filter with a slit width of 20 eV. One dataset of Cx43-WT in GDN at pH 8.0 was collected at KAIST Analysis center for Research Advancement (KARA, Daejeon, Korea) using a Glacios microscope (Thermo Fisher Scientific, USA) at 200 kV equipped with a Falcon 3EC detector. All the detectors were operated in electron counting mode. Data collections were carried out using a EPU automatic data acquisition software (Thermo Fisher Scientific, USA). Detailed data acquisition conditions and parameters are provided in Supplementary Table 1.

### Cryo-EM data processing

All datasets of Cx43 samples were processed with RELION 3.1[40] or cryoSPARC version 3.1[41] software in a similar way. Beam-induced motion correction and dose weighting of raw movies were performed using MotionCor2 version 1.2.6[42] or Patch motion correction. Contrast transfer function (CTF) estimation was performed using Gctf version

1.18[43] or Patch CTF estimation. Micrographs unsuitable for further image processing such as those containing ice crystals, large motion drifts, very low or high defocus, and large astigmatisms were removed by manual inspection in Selection job in RELION or Manually Curate Exposures job in cryoSPARC. Particles were semiautomatically picked using Auto-picking or Template picker with the initial 2D templates, which were generated by randomly selected 500 micrographs. Many false-positive images, poorly aligned particles, off-centered particles, and undocked hemichannel particles were discarded by performing about 10 rounds of two-dimensional (2D) classification. An ab initio 3D reference model of Cx43 GJICh was generated by a stochastic gradient descent algorithm in RELION with C1 symmetry. After several rounds of 3D classification with C1 or D6 symmetry imposition, particles in the classes with good-quality side views of GJIChs were used in further 3D refinement. For further improvement of the map, per-particle motion drifts of assorted particles were corrected using Bayesian polishing[44] and per-particle CTF estimation was performed using CTF refinement[45] in some cases.

For all datasets, the resolution of the final map was estimated based on the 0.143 Fourier shell correlation (FSC) criterion[46]. The local resolution of the density map was assessed by ResMap[47] (Supplementary Fig. 4). Detailed processing information including map symmetry, number of particle images, and map resolution, etc. are summarized in Supplementary Table 1.

### Protomer-focused classification and statistical analysis on the relationship between protomers in GJIChs

In the datasets of Cx43-WT in LMNG/CHS at pH 6.9 and Cx43-M257 both in LMNG/CHS and in nanodiscs (POPE) at pH 8.0, during 3D classification steps with C1 symmetry, we identified two noticeable classes with and without NTH densities. To distinguish possible different classes representing various conformations of NTHs, we used single protomer-based conformational variation analysis[48]. For this analysis, 3.5 Å map of Cx43-WT in LMNG/CHS at pH 6.9, 3.3 Å map of Cx43-M257 in LMNG/CHS at pH 8.0, and 3.5 Å map of Cx43-M257 in nanodiscs (POPE) at pH 8.0 were used as consensus maps. Using already well-aligned particles used for reconstructing the consensus maps, we ran a command "relion_particle_symmetry_expand" in RELION 3.1 to produce 11 copies of each particle by computationally rotating and tilting based on the D6 symmetry axis. This resulted in 12 times the total number of particles, which were subjected to Particle subtraction job to generate subtracted particles containing only the signal of one protomer. The subtracted particles were re-centered and re-extracted into 200 × 200 pixel boxes, and then subjected to a focused 3D classification with a mask covering one protomer and without orientation search. This resulted in two distinct classes of protomers with GCN and PLN conformations for both datasets of Cx43-WT in LMNG/CHS at pH 6.9 and Cx43-M257 in LMNG/CHS at pH 8.0, and three distinct classes of protomers with GCN, FIN and PLN conformations for dataset of Cx43-M257 in nanodiscs (POPE) at pH 8.0 (Supplementary Fig. 3a, b).

To see the distribution of how many PLN protomers (protomers in the PLN conformation) are included and located in each GJICh particle, we analyzed the metadata files generated by focused 3D classification. The metadata contain all subtracted protomer particles labeled with the class number of each protomer, the angles by which subtracted protomer particles were rotated and tilted from original GJICh particles to find out their relative locations and the identification (ID) number of each original GJICh particle (refer to 'ImageOriginalName') to which each protomer belongs. Because of D6 symmetry expansion, the same identification number of GJICh should be found 12 times in the metadata. The metadata was sorted by the protomer class numbers from focused 3D classification, and the GJICh ID numbers were collected only for the classes with the PLN conformation. How many times the same ID is repeated in the collection indicates how many PLN

protomers are included in the corresponding GJICh particle. To sort particles in the metadata, count the number of repetitions of a GJICh ID, and calculate the number of GJIChs with each of 13 different protomer compositions (0:12 to 12:0 of GCN and PLN) at the same time, we used the custom-made shell script. In addition, by analyzing the relative locations of PLN and GCN protomers in each half of the GJICh particle in our experimental dataset and by using simple probability statistics as shown in a previous report[13], we further investigated whether a conformational change in each protomer seems to be in an independent or cooperative manner relative to adjacent protomers.

### Determination of hemichannel sub-structures with different NTH conformations by focused classification and localized reconstruction

The datasets of Cx43-M257 both in LMNG/CHS and in nanodiscs (POPE) at pH 8.0 were used for focused classification on and localized reconstruction of hemichannel regions. We followed the procedures reported by Gestaut et al.[49] and Zhao et al.[50]. The overall workflow is presented in Supplementary Fig. 3c. To cover the orientation of both hemichannels in each GJICh particle, we used two approaches by Gestaut et al.[49] and Zhao et al.[50] for the dataset of Cx43-M257 in LMNG/CHS and the dataset of Cx43-M257 in nanodiscs (POPE), respectively.

In the dataset of Cx43-M257 in LMNG/CHS, we used two masks covering each hemichannel of the consensus GJICh map to generate the subtracted particles in two opposite orientations. After re-centering and re-extraction into 300 × 300 pixel boxes, we subjected the subtracted particles to 3D classification ($K = 6$) with C1 symmetry imposition to align into one orientation. We chose five hemichannel sub-structure classes (Class 1, 3, 4, 5, and 6) with good quality side views and performed focused 3D refinement (C1 symmetry) with a soft mask covering hemichannel to obtain a hemichannel sub-structure map at 3.8 Å resolution. Using this map as a new consensus map, we ran 25 iterations of 3D classification ($K = 8$, $T = 4$; $T$ is Regularization parameter) without orientation search. In additional 15 iterations of 3D classification, we increased the $T$ value ($T = 40$) and applied a soft mask covering the cytoplasmic half of hemichannel sub-structures to focus more on the conformational changes of NTHs. This generated one unique class (Class 6) in the fully PLN state. 8,046 particles in Class 6 were subjected to 3D refinement with C1 symmetry imposition and 1.8° angular sampling, which produced a 7 Å map. Another round of focused 3D refinement with C6 symmetry imposition resulted in a 4.3 Å map. Next, we traced back the original GJICh particles from which the subtracted hemichannel particles came, and redundant particles were removed. Total 7,446 GJICh particles were subjected to 3D refinement with C6 symmetry imposition and 1.8° angular sampling. The result showed a 3.6 Å GJICh map with fully PLN hemichannel on one side and fully GCN hemichannel on the other side. This map was not subjected to sharpening because it significantly weakens map densities of NTHs. The local resolution of the density map was assessed by ResMap.

In the dataset of Cx43-M257 in nanodiscs (POPE), we ran a command "relion_particle_symmetry_expand" in RELION 3.1 with D1 symmetry to align both hemichannels in each GJICh particle into one orientation, producing one more copy of each particle. Then, we used a mask covering one hemichannel to generate the subtracted hemichannel particles. After re-centering and re-extraction into 256 × 256 pixel boxes, the subtracted hemichannel particles were subjected to 25 iterations of 3D classification ($K = 8$, $T = 4$) with C6 symmetry imposition without orientation search. In additional 15 iterations of 3D classification, we increased the $T$ value ($T = 10$) and applied a soft mask covering the hemichannel sub-structures. This generated three unique classes: Class 4 showing the fully PLN state, and Class5 and Class 6 not adopting the fully PLN state. To further classify, Classes 4, 5 and 6 were subjected to 3D classification in a similar manner ($K = 5$, $T = 10$ or up to 20). As a result, four distinguished classes of hemichannel sub-

structures (GCN, GCN[TMLi], FIN, and PLN state) were generated and these density maps were further improved with 3D refinement with C6 symmetry imposition and 1.8° angular sampling, resulting in 3.8 Å, 3.8 Å, 3.7 Å, and 4.0 Å map, respectively. These density maps were also not subjected to sharpening and the local resolution maps were calculated by ResMap.

## Model building, validation, and structural analysis
The structural model for the Cx43-WT GJICh in LMNG/CHS at pH 8.0 was built manually in Coot program[51,52] and refined using phenix.real_space_refine[53] in PHENIX software suite with secondary structure and non-crystallographic symmetry (NCS) restraints. This structure does not contain Met1 and N-terminal acetylation (see the main text for details). The final model also does not include flexible cytoplasmic regions such as CLs (residues 111–147) and CTDs (residues 238–382) because the map densities of these regions are invisible. We also put acyl chain models into long map densities around TMDs but did not put any lipid model into unidentified densities presumed to be a head group of phospholipids/detergents and a sterol because of ambiguous map densities. The geometric restraints for acyl chains were optimized using the eLBOW module[54] in PHENIX. For the models of Cx43-WT GJICh in GDN and in nanodiscs (soybean lipids) at pH 8.0, the Cx43-WT GJICh protein model in LMNG/CHS at pH 8.0 was fitted into the map densities using Chimera's Fit in Map tool, and real-space refinement were performed in PHENIX with secondary structure and NCS restraints. For the models of Cx43-WT GJICh in nanodiscs (POPE/CHS) and Cx43-M257 GJICh in nanodiscs (POPE/CHS) at pH 8.0, model building was performed in a similar manner as mentioned above, except for lipid models. Considering the resolution and quality of maps, we could build lipid models (CHS and POPE) into map densities of which geometric constraints were generated using the eLBOW module. For the models of Cx43-M257 hemichannel (half of GJICh) in nanodiscs (POPE) at pH 8.0, especially in FIN conformation, well modeled NTH in GCN conformation was manually tilted in Coot and refined over multiple rounds using real-space refinement in PHENIX.

For the structure of Cx43-M257 GJICh in LMNG/CHS at pH 8.0 (C6 symmetry) composed of a PLN and a GCN hemichannel, we first performed homology modeling in the *SWISS-MODEL* server[55] using an available Cx46 structure (PDB code number 7JKC)[8] as a template. The predicted Cx43 model was fitted into one PLN protomer map of the Cx43-M257 GJICh map in Chimera, manually modified in Coot, and refined in PHENIX. The refined model of one protomer in the PLN conformation was fitted into the five other PLN protomer maps. For modeling the GCN hemichannel part, a hemichannel model from the Cx43-WT GJICh structure in LMNG/CHS at pH 8.0 was fitted into the map in Chimera, and then the dodecameric model was refined in PHENIX (real-space refinement) with secondary structure and NCS restraints.

For validation of structural models, FSC curves were calculated between EM maps and final models. The qualities of the final models were evaluated using MolProbity[56] in PHENIX. Detailed statistics for model refinement and validation are presented in Supplementary Table 1. The pore radii and the surface of the pore pathway were calculated using HOLE[57] and CAVER v.3.03[58]. Figures were produced with Chimera and Pymol.

## MD simulation protocol
For all MD simulations, we used the GROMACS 2020.2 package[59] and the CHARMM36m force field[60], combined with the CHARMM-modified TIP3P model and the CUFIX corrections for charge-charge interaction pairs[61]. All simulations were performed under a constant surface tension−constant temperature (NPγT) ensemble at zero surface tension (γ = 0) using the Parrinello-Rahman scheme[62] and 300 K temperature using the Nose−Hoover scheme[63]. Van der Waals forces were computed using a 10- to 12-Å switching scheme. Long-range electrostatic forces were computed using the particle-mesh Ewald summation scheme[64] of a 1.2-Å grid spacing and 12-Å real-space cutoff. Covalent bonds to hydrogen in non-water and water molecules were constrained using the LINCS[65] and SETTLE[66] algorithms, respectively. During MD simulations using a 2-fs time step, we saved atomic coordinates every 20 ps for analysis. The visualization scheme for the density-flux map and the computation scheme for the ionic current is described in a previous report by Yoo and Aksimentiev[33]. To estimate the statistical error of the computed ionic currents, we divided the 200-ns trajectory into four non-overlapping 50-ns blocks and computed the standard error of four block averages. It should be noted that simulated currents may need to be scaled by a factor of three because the viscosity of the CHARMM-modified TIP3P water model is about a third of the experimental value[67].

## MD preparation and simulation of Cx43 GJICh embedded in a lipid bilayer
We manually reconstructed the unstructured CL of each Cx43 chain (residues 111–147) missing in experimental GJICh models; the reconstructed loops were in a random configuration with no clash and no intertwining between chains. To create a double bilayer system, we placed a lipid bilayer of a 1-palmitoyl-2-oleoyl-*sn*-glycero-3-phosphatidylcholine (POPC) near the transmembrane domain of each hemichannel and removed lipid molecules overlapping with the channel. We immersed the complex system of lipid molecules and the channel in an explicit solution of 150 mM NaCl. The final system contained a channel, 734 POPC lipids, 88,000 water molecules, 300 Na ions, and 360 Cl ions in a periodic hexagonal box ($a = b \approx 14$ nm, $c \approx 26$ nm, $\alpha = \beta = 90°$, $\gamma = 60°$). We simulated each of the assembled systems for 400 ns in total (200 ns at a voltage bias of 0 mV with position restraints on the experimentally determined heavy atoms followed by 200 ns at a voltage bias of 200 mV). We energy-minimized each system for 5000 steps and equilibrated it for 200 ns with the positions of experimentally determined heavy atoms harmonically constrained using a force constant of 1000 kJ/mol under zero voltage bias; note that we did not constrain the positions of reconstructed CLs. For the measurements of ionic currents, we performed a 200-ns simulation under 200 mV without any constraints for each system, starting from the final structure of the equilibration; we estimated the error as the standard error of four 50-ns block averages by dividing the 200-ns trajectory into four non-overlapping trajectory blocks. It should be noted that we determined the magnitude of electric fields by dividing the magnitude of electric potential by the height of the box (Lz) as suggested by the Roux group[68].

The root mean square deviations (RMSDs) of the full PLN GJICh, the 3:3 mixed GJICh, and the FIN hemichannel sub-structure converged to about ~2.8 Å, ~2.9 Å, and ~4.3 Å, respectively, suggesting that these channels remained structurally stable under a thermal fluctuation (Supplementary Fig. 10f). To see whether there exist particularly dynamic structural motifs, we computed the root mean square fluctuation (RMSF) using the trajectory more than 200-ns (Supplementary Fig. 10g). The results for the full PLN GJICh and the 3:3 mixed GJICh showed the RMSF values of ~1 Å for all residues, except those in six CLs and several terminal residues, suggesting that these GJICh structures are stable in lipid bilayers. Compared with these two GJIChs, the FIN hemichannel sub-structure showed substantial dynamicity (RMSF of ~2.3 Å) in the region spanning NTH and the NTH-TM1 loop. In addition, the other two regions in ECLs (around the residues 60 and 190) showed slight increase of dynamicity probably due to the absence of the interconnexon docking interaction.

## Reporting summary
Further information on research design is available in the Nature Portfolio Reporting Summary linked to this article.

## Data availability

All data needed to evaluate the conclusions in the paper are present in the paper and/or the Supplementary Information. Atomic coordinates and cryo-EM density maps have been deposited in PDB and EMDB as follows: Cx43-WT GJICh (pH 8.0, LMNG/CHS) in the full GCN conformation, PDB 7F92 and EMD-31495; Cx43-M257 GJICh (pH 8.0, LMNG/CHS) with two conformationally different hemichannels, PDB 7F94 and EMD-31497; Cx43-WT GJICh (pH 8.0, GDN) in the full GCN conformation, PDB 7XQ9 and EMD-33391; Cx43-WT GJICh (pH 8.0, Soybean lipids) in the full GCN conformation, PDB 7F93 and EMD-31496; Cx43-WT GJICh (pH 8.0, POPE/CHS) in the full GCN conformation, PDB 7XQB and EMD-33392; Cx43-M257 GJICh (pH 8.0, POPE/CHS) in the full GCN conformation, PDB 7XQF and EMD-33394; Cx43-M257 GJICh (pH 8.0, POPE/CHS, C1 symmetry) in the full GCN conformation, PDB 7XQD and EMD-33393; The hemichannel region of Cx43-M257 GJICh (pH 8.0, POPE) in the full GCN conformation, PDB 7XQG and EMD-33395; the hemichannel region of Cx43-M257 GJICh (pH 8.0, POPE) in the full GCN$^{TMIi}$ conformation, PDB 7XQH and EMD-33396; the hemichannel region of Cx43-M257 GJICh (pH 8.0, POPE) in the full FIN conformation, PDB 7XQI and EMD-33397; the hemichannel region of Cx43-M257 GJICh (pH 8.0, POPE) in the full PLN conformation, PDB 7XQJ and EMD-33398. The consensus cryo-EM density map of Cx43-WT GJICh (pH 6.9, LMNG/CHS) has been deposited in EMDB with ID EMD-33399. Source data are provided as a Source Data file. Source data are provided with this paper.

## Code availability

MD trajectories have been deposited to Zenodo (https://doi.org/10.5281/zenodo.7219679).

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

## Acknowledgements

The authors thank Bumhan Ryu at IBS and Jin-Seok Choi at KARA for their assistance in data collection. We also thank Amer Alam at the University of Minnesota for critical reading of the manuscript, and Jeesoo Kim and Jong-Seo Kim at the Seoul National University for mass spectrometry experiments and data analyses. This work was supported by the Suh Kyungbae Foundation (SUHF-18010097 to J.-S.W.), the National Research Foundation (NRF) grants funded by the Ministry of Science and ICT (NRF-2018R1C1B6004447 to J.-S.W. and NRF-2020R1A2C1101424 to J.Y.), and by the National Supercomputing Center (KSC-2020-CRE-0080 to J.Y.).

## Author contributions

J.-S.W. conceived this project; H.-J.L., H.J.C., S.-N.L., and C.-W.L. purified GJICHs; H.-J.L and H.J. performed electron microscopy; H.-J.L., H.J.C., and H.J. analyzed the EM data and determined the structures; M.K. and J.Y. performed MD simulations; H.-J.L., H.J.C., H.J., and J.-S.W. wrote the manuscript.

## Competing interests

The authors declare no competing interests.
