## [Peer Review File · Nature Communications]

Conformational changes in the human Cx43/GJA1 gap junction channel visualized using cryo-EMEditorial Note: This manuscript has been previously reviewed at another journal that is not operating a transparent peer review scheme. This document only contains reviewer comments and rebuttal letters for versions considered at *Nature Communications*.

REVIEWERS' COMMENTS

Reviewer #1 (Remarks to the Author):

Since the last version, the authors have made additional revisions to make it more comprehensible for readers.

The authors have addressed all my concerns.

I have no further comments.

Reviewer #2 (Remarks to the Author):

The revised work by Lee et al. reflects a careful and attentive response to my previous critiques. I have no further significant concerns. The authors are congratulated on their efforts to characterize this important system!

I do provide a couple minor comments below that the authors might consider in preparing a final submission, which should be readily addressed without further review.

Kind regards –

Minor

Line 250: "...indicating that the heads are more ordered than the tails." May be more appropriate to state "...indicating that the heads are more ordered than the tails and/or display enhanced electron scattering by the composition of heavier atoms."

Line 259: "..., and thus the central pore formed by..." consider changing to the following for improved clarity "..., and thus the cytoplasmic end of the central pore formed by..."

Line 274: Ideally, the authors would state what are the quantitative differences in [CHS] under these two conditions in the main text.

Reviewer #3 (Remarks to the Author):

Authors had addressed properly all my previous concerns. After carefully reading the new manuscript version, I just have a few minor comments.

Minor comments:

1. It is not clear the pink cylinder representing the NT in Fig. 6C (FIN conformation). This representation is confusing since the other panels in Fig. 6C (for GCN and PLN) only show 1 NT instead of two and they are not represented as a cylinder. This is not explained in figure legend neither any part of the manuscript.

2. Related to the edited abstract:

- Lines 38-39: "The conformational equilibrium shifted to GCN by cholesteryl hemisuccinates and to PLN by C-terminal truncations.". This statement is not fully accurate since PLN conformation was also observed without CT deletion (i.e., at PH=6.9).

- Lines 41-43: "We observed an α -to- π -helix transition in the first transmembrane helix, which created a side opening to the membrane in the FIN conformation." Opening was also observed for PLN conformation.

3. Please specify in legend of Figure 3 what condition is shown.

4. Lines 326-328 "However, this equilibrium state of Cx43-WT GJICs in condition 3 would not represent the maximally open state of the channels at pH 6.9 in the previous electrophysiological experiments." Please incorporate the reference to the paper with electrophysiological experiments.

Responses to Reviewers

We thank the reviewers for their time and effort to thoroughly read our point-by-point response and the revised manuscript and give us minor comments to increase the quality of the manuscript. All changes in the text have been highlighted by blue letters.

Reviewer #1

Reviewer #1 had no further comments.

Reviewer #2

Minor points

(1) Line 250: "...indicating that the heads are more ordered than the tails." May be more appropriate to state "...indicating that the heads are more ordered than the tails and/or display enhanced electron scattering by the composition of heavier atoms."

We agree with the reviewer and have modified the sentence according to the reviewer's suggestion (Line 248).

(2) Line 259: "..., and thus the central pore formed by..." consider changing to the following for improved clarity "..., and thus the cytoplasmic end of the central pore formed by..."

We agree with the reviewer and have modified the sentence according to the reviewer's suggestion (Line 258).

(3) Line 274: Ideally, the authors would state what are the quantitative differences in [CHS] under these two conditions in the main text.

As suggested by the reviewer, we have added approximate concentrations (mol%) of CHS in the nanodisc samples under condition 7 and 8. The corresponding sentences have been modified as following.

"Because the protein sample included only 0.0005% (w/v) CHS, the lipid-protein mixture contained approximately 300 fold more phospholipid molecules than CHS molecules. In addition, CHS as well as LMNG would be partly removed during the incubation with adsorbent beads. Therefore, the CHS content in the lipid nanodisc sample was likely

less than 0.3 mol%, which should be much lower than that under condition 7 (less than 50 mol%).” (Line 271-276)

Reviewer #3

Minor points

(1) *It is not clear the pink cylinder representing the NT in Fig. 6C (FIN conformation). This representation is confusing since the other panels in Fig. 6C (for GCN and PLN) only show 1 NT instead of two and they are not represented as a cylinder. This is not explained in figure legend neither any part of the manuscript.*

Following the reviewer's suggestion, we have added an appropriate explanation in the Figure 6C legend.

(2)-1 *Lines 38-39: “The conformational equilibrium shifted to GCN by cholesteryl hemisuccinates and to PLN by C-terminal truncations.”. This statement is not fully accurate since PLN conformation was also observed without CT deletion (i.e., at PH=6.9).*

We agree with the reviewer and have modified the sentence as the editor suggested.

“The conformational equilibrium shifts to GCN by cholesteryl hemisuccinates and to PLN by C-terminal truncations and at varying pH.” (Line 38)

(2)-2 *Lines 41-43: “We observed an α -to- π -helix transition in the first transmembrane helix, which created a side opening to the membrane in the FIN conformation.” Opening was also observed for PLN conformation.*

We agree with the reviewer and have added “the PLN conformation” in the sentence (Line 43).

(3) *Please specify in legend of Figure 3 what condition is shown.*

We have added the explanation in the Figure 3 legend.

(4) Lines 326-328 “However, this equilibrium state of Cx43-WT GJICs in condition 3 would not represent the maximally open state of the channels at pH 6.9 in the previous electrophysiological experiments.” Please incorporate the reference to the paper with electrophysiological experiments.

Following the reviewer's suggestion, we have referenced the paper.

Reviewer #4

Minor points (Technical details)

(1) Details on P- and T-coupling (algorithm and coupling constant) are missing.

To control the pressure and temperature of systems, we used the Parrinello-Rahman (Ref. 62 in the manuscript) and the Nose-Hoover schemes (Ref. 63 in the manuscript), respectively. Along with the references, we explicitly specified those schemes in the revised manuscript.

Line 946: “at zero surface tension ($\gamma = 0$)⁶² and 300 K temperature⁶³” to “at zero surface tension ($\gamma = 0$) using the Parrinello-Rahman scheme⁶² and 300 K temperature using the Nose-Hoover scheme⁶³”

(2) Are you sure that you constrained water molecules with LINCS, which is rather uncommon? The use of SETTLE is more common for water molecules with Gromacs.

Although we used the SETTLE algorithm for water molecules and correctly cited the original SETTLE paper (Ref. 66 in the manuscript), SETTLE was not explicitly stated in the original manuscript. We have added the following statement in the revised manuscript.

Line 951: “using the LINCS⁶⁵ and algorithms⁶⁶, respectively.” to “using the LINCS⁶⁵ and SETTLE⁶⁶ algorithms, respectively.”

(3) The Charmm-modified TIP3P water model (as the normal TIP3P model) exhibits a too large diffusion coefficient, which may lead to increased ionic currents relative to experiments. This limitation does not affect the main conclusions of this article but may affect the qualitative results of the simulations. The authors may either mention and

discuss this limitation or carry out a control simulation with a different water model with correct diffusion coefficient (such as the OPC, TIP4P/2005, or TIP3P-force-balance). I would leave the decision to the authors, either solution is fine.

As pointed out by the reviewer, it is well-known that the TIP3P water model underestimates the viscosity by a factor of three (Ref. Lee et al.). As mentioned by the reviewer, we believe that this deviation does not affect the main conclusion of the manuscript because the effect of viscosity on current measurements can be removed by scaling the currents by a constant factor. As suggested by the reviewer, we have added the following sentence in Methods section.

“It should be noted that simulated currents may need to be scaled by a factor of three because the viscosity of the CHARMM-modified TIP3P water model is about a third of the experimental value (Ref. Lee et al.); in the manuscript, we did not scale the current values.” (Line 956)

Ref. Lee, H.; Venable, R. M.; MacKerell, A. D., Jr; Pastor, R. W. Molecular Dynamics Studies of Polyethylene Oxide and Polyethylene Glycol: Hydrodynamic Radius and Shape Anisotropy. *Biophys. J.* 2008, 95 (4), 1590–1599.

*(4) How was the voltage of 200mV implemented? Using an external electric field E_z following $200\text{mV} = E_z * L_z$? Please provide more details.*

Yes, we determined the magnitude of electric fields by dividing the magnitude of electric potential by the height of the box (L_z) as suggested by the Roux group (Ref. DOI: <https://doi.org/10.1016/j.bbamem.2011.09.030>). As suggested by the reviewer, we have added the following sentence in Methods section.

“It should be noted that we determined the magnitude of electric fields by dividing the magnitude of electric potential by the height of the box (L_z) as suggested by the Roux group (Ref. DOI: <https://doi.org/10.1016/j.bbamem.2011.09.030>).” (Line 978)

(5) Line 164: Did you mean Fig. 1d instead of Fig. 1c?

Yes. We have corrected the error.

(6) The Zenodo archive at <https://zenodo.org/record/7219679> contains only structures

and trajectories. For reproducibility, topologies, MD parameter (mdp), and (if applicable) index files should be added for all systems.

As suggested by the reviewer, we have added topologies, MD parameter (mdp) files at <https://zenodo.org/record/7219679>.